# Enhancing the sensitivity of atom-interferometric inertial sensors using robust control

Jack C. Saywell[1], Max S. Carey[1], Philip S. Light[1], Stuart S. Szigeti [1], Alistair R. Milne[1], Karandeep S. Gill[1], Matthew L. Goh[1], Viktor S. Perunicic[1], Nathanial M. Wilson[1], Calum D. Macrae[1], Alexander Rischka [1], Patrick J. Everitt[1], Nicholas P. Robins[1], Russell P. Anderson[1] ✉, Michael R. Hush[1] & Michael J. Biercuk[1]

Atom-interferometric quantum sensors could revolutionize navigation, civil engineering, and Earth observation. However, operation in real-world environments is challenging due to external interference, platform noise, and constraints on size, weight, and power. Here we experimentally demonstrate that tailored light pulses designed using robust control techniques mitigate significant error sources in an atom-interferometric accelerometer. To mimic the effect of unpredictable lateral platform motion, we apply laser-intensity noise that varies up to 20% from pulse-to-pulse. Our robust control solution maintains performant sensing, while the utility of conventional pulses collapses. By measuring local gravity, we show that our robust pulses preserve interferometer scale factor and improve measurement precision by 10× in the presence of this noise. We further validate these enhancements by measuring applied accelerations over a 200 $\mu g$ range up to 21× more precisely at the highest applied noise level. Our demonstration provides a pathway to improved atom-interferometric inertial sensing in real-world settings.

Quantum sensors based on light-pulse atom interferometry have performed inertial measurements of unparalleled sensitivity and stability in laboratory environments[1-8]. However, there exist significant challenges to adapting such sensors for field-based operation[9,10]. Operation on a moving platform – necessary for deployment on a ship, aircraft, or spacecraft – typically degrades measurement sensitivity by many orders of magnitude[11-14] due to a variety of physical mechanisms. One example mechanism is rotation-induced Coriolis phase shifts[15], which can be mitigated through established hardware solutions such as tip-tilt mirrors[16] and active gyro-stabilization platforms[13]. Another example is variations in the laser intensity experienced by the atoms arising from relative motion between the free-falling atoms and the fixed laser beams[12,17], which cause errors in the atom-light coupling that degrade the quality of the beamsplitters and mirrors. Errors also arise due to the momentum distribution of the atomic source, since a spread in the Doppler-detunings away from resonance gives a spread

of atom-light couplings amongst the atomic sample. Although this can be mitigated by selecting only a narrow fraction of atoms from a broad thermal momentum distribution[11], this substantially reduces the measurement signal-to-noise ratio.

One solution to these challenges is the adoption of light pulses that, by design, make the interferometer highly resilient to noise. For example, conventional composite and adiabatic pulses designed for nuclear magnetic resonance (NMR)[18,19] have been deployed alongside conventional matter-wave beamsplitters and mirrors and shown to increase an interferometer's space-time area, and therefore sensitivity[20-22]. However, this approach relies on delicate phase cancellation between pulse pairs, making it susceptible to new sources of failure. Alternatively, optimal and robust quantum control[23-26] may be used to design error-robust pulses that are tailored to the specific noise sources that afflict atom interferometers. The application of these tailored error-robust pulses has been explored primarily through

[1]Q-CTRL, Sydney, NSW, Australia. ✉e-mail: russell.anderson@q-ctrl.com

theoretical proposals[27–32]. Notable exceptions include refs. [33] and [34], which respectively deploy composite Floquet pulses and tailored Raman pulses to improve the pulse fidelity and fringe contrast in Mach−Zehnder interferometers.

Despite the strong level of interest, the lack of two critical demonstrations has prevented the wider uptake of error-robust atom interferometry. Firstly, an inertial measurement has never been made − let alone improved − by a cold-atom quantum sensor that employs tailored error-robust light pulses, leaving the fundamental utility of this approach unresolved. Secondly, the interferometer scale factor, which relates an acceleration of interest to the measured phase shift, has never been quantified for error-robust light-pulse sequences, nor confirmed to be stable under noisy operating conditions. Maintaining a known, stable scale factor is crucial for preserving the key advantage of quantum cold-atom sensors: their accuracy and long-term stability.

In this work, we provide these critical experimental demonstrations and thus establish error-robust control at the software layer as a viable solution to key challenges in field-deployed cold-atom inertial sensing. We develop error-robust Bragg beamsplitters and mirrors that are resilient to variations in the atom-light coupling strength and detuning, thereby enabling robustness to broad atomic source momentum distributions, cloud expansion, and uncontrolled atomic motion due to platform accelerations transverse to the propagation direction of the interferometry beams. We deploy these pulses on a state-of-the-art cold-atom accelerometer with a broad momentum width atomic source (~1.6 ℏk) and confirm that they yield the expected scale factor through a measurement of local gravity that is 2× more precise than a gravitational measurement made using conventional Gaussian pulse sequences. In the presence of laser intensity noise that varies up to 20% from pulse-to-pulse (chosen to emulate lateral motion caused by platform accelerations ~1g in size acting during a 10 ms interferometer), interferometry with Gaussian pulses is no longer performant, losing precision and becoming inaccurate. In contrast, our error-robust pulses maintain the accuracy and precision of the interferometer's phase measurements, delivering a 10× relative improvement in phase-estimation uncertainty compared to Gaussian pulse sequences. We validate this laser-noise resilience in a direct measurement of applied platform accelerations over a 200 μg range, with our error-robust pulses yielding a stable scale factor in the presence of 20% laser intensity noise and up to 21× improvement to measurement precision over Gaussian pulses.

## Results

### Error-robust Bragg pulses for atom interferometry

We consider a standard three-pulse ($\pi/2$-$\pi$-$\pi/2$) Mach−Zehnder interferometer sequence where multi-photon Bragg pulses[35,36] are used to effect beamsplitting and reflection of the atomic matter-waves (see Fig. 1a). Our interferometer (described in the Methods) employs counter-propagating vertical Bragg beams in a retroreflecting arrangement which are detuned ~8 GHz above the $F = 1 \rightarrow F' = 2$ transition of the [87]Rb D2 line. The wavenumber $k$ of these counter-propagating beams is approximately equal such that, in the atomic rest frame, they are separated in frequency only by an integer multiple $n$ of the two-photon recoil frequency[37]. This drives $2n$-photon transitions between the same electronic state, allowing the coherent coupling of states separated in momentum by $2n\hbar k$. This coupling is parameterized by the laser frequency difference (two-photon detuning) $\delta$ and a complex two-photon Rabi frequency with amplitude $\Omega_R$ and phase $\phi_L$ (see Methods), where $\Omega_R$ is proportional to the laser intensity and $\phi_L$ is the relative optical phase between the two beams. Our laser system enables precise control over pulse parameters ($\Omega_R, \delta, \phi_L$) via the optical intensity, relative phase and frequency of the two Bragg beam frequency components − which is crucial to execute our tailored error-robust interferometry pulses.

Conventional Bragg-pulse atom interferometers use pulses whose temporal amplitude envelope is Gaussian[36]. In these conventional pulses the relative phase $\phi_L$ and frequency $\delta$ of the lasers is fixed (in the inertial

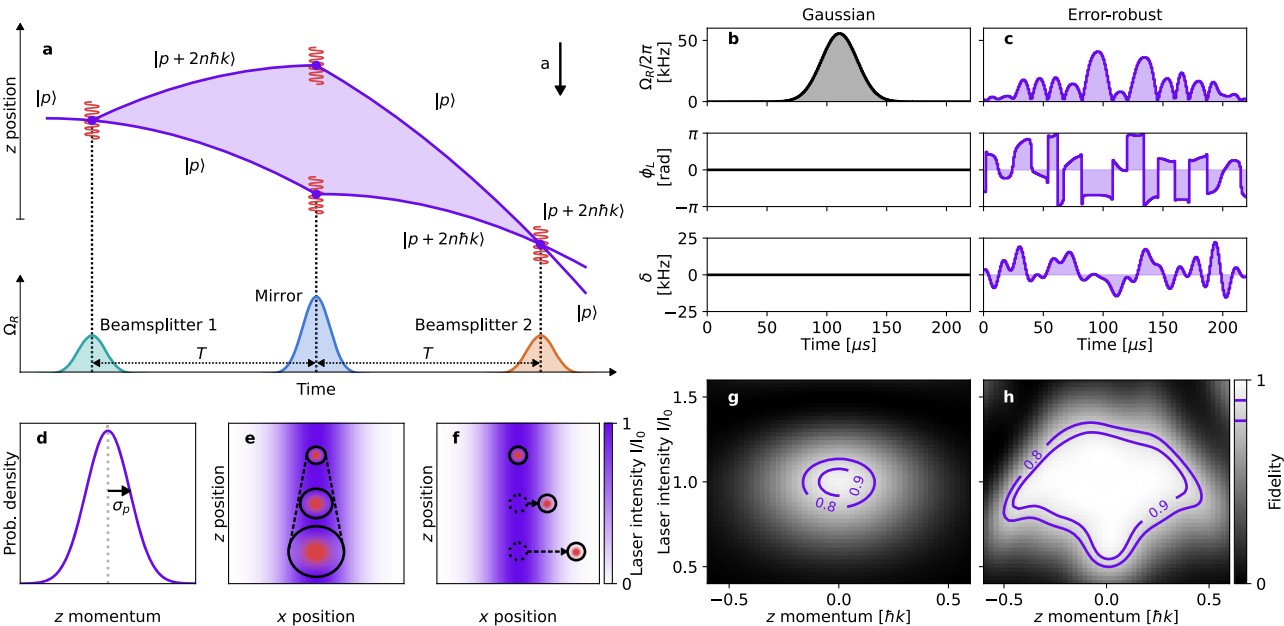

**Fig. 1 | Sources of noise in field-deployed atom-interferometric sensors and error-suppression using error-robust pulses. a** Space-time diagram for an order-$n$ Bragg pulse atom interferometer in the Mach−Zehnder configuration, employing conventional Gaussian pulses as mirror and beamsplitters separated by equal interrogation times $T$ and subject to a constant acceleration $a$ in the $-z$ direction. **b** and **c** show the waveforms for Gaussian ($\sigma_\tau = 15\,\mu s$) and error-robust Bragg mirror pulses of order-3 (6 ℏk), respectively. (d-f) show schematic representations of the relevant noise processes under consideration (see main text). Noise source: **d** the finite atomic momentum width of the atomic source in the longitudinal $z$ direction; **e** the thermal expansion of the atom cloud in the transverse $xy$ plane across a Gaussian beam; and **f** the effect of a constant platform acceleration transverse to the measurement axis. The state-transfer fidelity of the Gaussian and error-robust mirror pulses as a function of atomic $z$ momentum and laser intensity variation (plotted as a fraction of the peak intensity $I_0$) is depicted in **g** and **h**, respectively, highlighting the improvement afforded by the error-robust pulse.

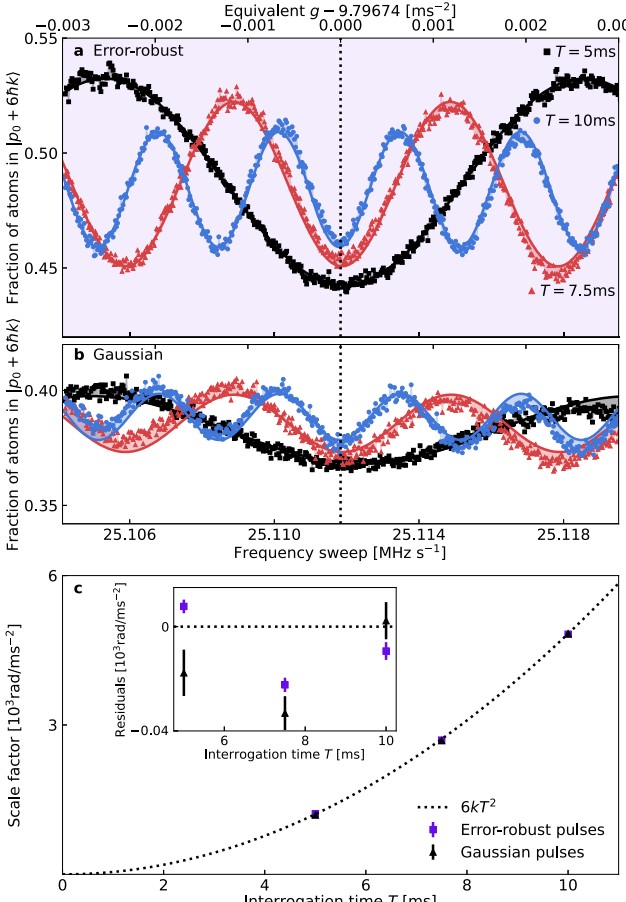

**Fig. 2 | Verifying the scale factor of error-robust Bragg interferometry by measuring gravity.** Interference fringes for (**a**) error-robust (highlighted by purple shading) and (**b**) conventional Gaussian order-3 pulse sequences, obtained by varying the sweep rate of the laser frequency difference for three different interrogation times; sinusoidal fits to each fringe are shown by solid lines and the regions between all fringes and corresponding sinusoidal fits are shaded to highlight the linear trend in the Gaussian fringes. **c** The measured scale factor, determined from the frequency of sinusoidal fits to the data in **a** and **b** with a linear trend, as a function of interrogation time $T$. Error bars denote $\pm 1$ standard error in the frequencies (scale factors) obtained from sinusoidal fits to each fringe. The dotted line plots the theoretically expected scale factor $6kT^2$ (ignoring fractional corrections of order $(\Omega_{max}T)^{-1}$[41]), residuals from which are displayed on the inset axes.

frame of the atoms), and the beam intensities are varied such that

$$\Omega_R(t) = \Omega_{max} \exp\left[-t^2/(2\sigma_\tau)^2\right], \qquad (1)$$

where $\Omega_{max}$ is the pulse amplitude (peak two-photon Rabi frequency) and the standard deviation $\sigma_\tau$ defines the pulse width. However, Gaussian Bragg pulses are highly sensitive to variations in initial atomic momentum $p$ and laser intensity $I$[38,39]. This means the fidelity of a given pulse (how well it performs its role as a matter-wave beamsplitter or mirror) decreases as $p$ and $I$ are varied away from optimal calibration settings (see Fig. 1g), consequently reducing the interference fringe visibility and single-shot sensitivity of the device. Figure 1d–f depicts three key sources of error that lead to variations in $p$ and $I$ for a single-axis atom interferometer operated in a dynamic environment: the finite momentum width (defined as the $2\sigma$ width) of the typically Gaussian atomic momentum distribution along the measurement axis defined by the interferometry beams ($z$ or longitudinal direction); the thermal expansion of the atomic cloud in the $xy$ (transverse) plane; and transverse displacement due to platform accelerations in the $xy$ plane.

Through the numerical optimization procedure outlined in the Methods, we develop tailored light pulses, henceforth referred to as error-robust pulses, that can replace conventional Gaussian mirrors and beamsplitters and provide robustness to variations in atomic momentum and laser intensity, offsetting the above-mentioned performance degradation. Figure 1c shows the piecewise-constant waveform for our error-robust order-3 mirror pulse, which comprises 220 time-steps each of duration 1 µs. We validate its improved robustness compared with a Gaussian mirror of equivalent Bragg order through numerical simulation (see Methods) of the mirror state-transfer fidelity $|\langle p + 6\hbar k|\hat{U}_M|p\rangle|^2$ as $p$ and $I$ are varied from zero and $I_0$, respectively, where $I_0$ is the peak laser intensity. These calculations indicate a wide regime of high pulse fidelity for the error-robust pulse, centered on the central point of zero added noise; the contour of 90% fidelity extends over a range approximately 5 × wider than for the Gaussian pulse. This complex shape with a broad high-fidelity pedestal is characteristic of a control solution robust to the two target noise sources.

## Measurement of local gravity and scale factor

As a first experimental test, we verify the measurement scale factor given by our error-robust Bragg pulses by measuring Earth's gravitational field. We operate the error-robust pulses at a peak two-photon Rabi frequency of $\Omega_{max} = 2\pi \times 40$ kHz − the value for which they were designed − with no additional calibration, and compare them to order-3 Bragg pulses with a Gaussian profile given by Eq. (1), where $\sigma_\tau$ is fixed at 25 µs and the amplitude of each pulse is varied to tune the pulse area so as to maximize contrast. Figure 2a, b shows interference fringes for both the error-robust and Gaussian order-3 Bragg pulses obtained by scanning the chirp rate $\alpha$ of the laser frequency difference for interrogation times of 5, 7.5, and 10 ms. By overlapping the fringes for different values of $T$, we can identify a minimum common to all fringes where the laser frequency chirp rate exactly compensates the Doppler shift due to Earth's gravity. This provides a measurement of $g$, which is depicted by the vertical dotted line.

Upon fitting to these data we observe that all fringes share a common center to within 6 µ$g$ and that, compared to conventional Gaussian pulses, the error-robust pulses yield a 2.5–3.2 × improvement in interferometer fringe visibility for all values of $T$. The enhanced fringe visibility indicates improved overall sensitivity, which we attribute primarily to the improved velocity acceptance of the robust pulses given the broad longitudinal momentum width (-1.6 $\hbar k$) of our experiment's thermal atomic source, while the common center indicates that any additional bias introduced by the error-robust pulses is no greater than the 6 µ$g$ range of imputed fringe centers for these data.

Using error-robust pulses also reduces systematics that are present in these chirp-domain interference fringes. We observe a linear trend in the fringe offset atop the sinusoidal oscillations, highlighted by the shaded regions between the fringes and sinusoidal fits in Fig. 2b. If unaccounted for during least-squares regression, this trend shifts the location of the imputed fringe center increasingly as $T$ is reduced. Using error-robust pulses reduces the slope of the linear trend by at least 2.5× for all values of $T$ shown here. This phenomenon has been observed in Bragg pulse interferometers[35] and to our knowledge its origin is yet to be identified.

We posit that the linear trend may be related to the pulses being off-resonant when the sweep rate does not perfectly match the Doppler shift caused by longitudinal acceleration. Consequently, the velocity distribution excited by the laser pulses varies with the sweep rate, more so for Gaussian pulses than for error-robust pulses, which are less sensitive to Doppler shifts. Further investigation is required to fully characterize this effect, which could be detrimental when operating in a dynamic environment employing mid-fringe locking techniques[40].

When augmenting the sinusoidal model with a linear term, the value of $g$ extracted from the location of the central minimum for all

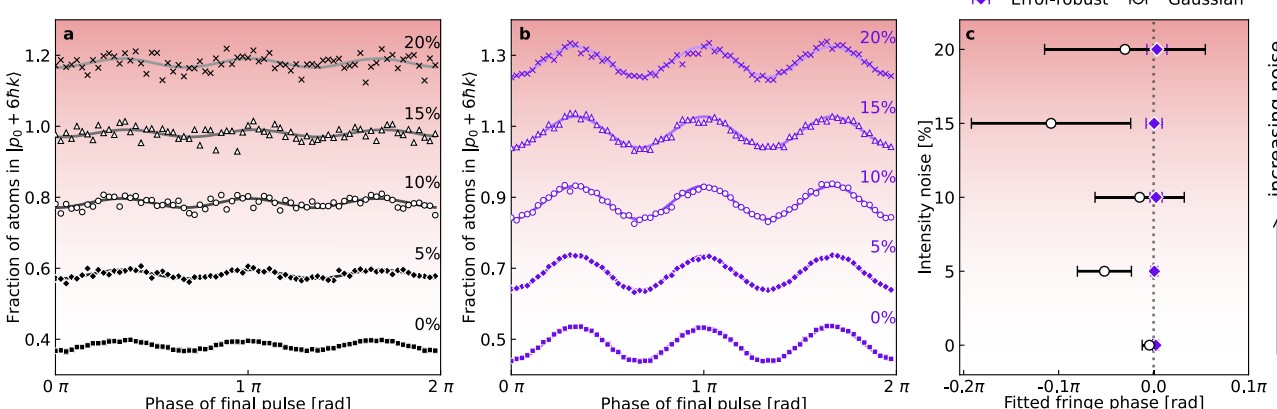

**Fig. 3 | Suppressing laser intensity noise using error-robust Bragg pulses.** Experimental fringes for $T = 5$ ms order-3 Bragg interferometers implemented using (**a**) conventional Gaussian pulses with a standard deviation $\sigma_\tau$ of 25 μs and (**b**) our error-robust pulses. The lowest, noise-free curves show actual measured population; additional fringes are offset vertically with a spacing of 20% for increasing values of applied noise $\sigma_\beta$ (labels in-set). The background color gradient is indicative of the level of applied noise. Fringes were recorded with the interferometer running in this configuration for a range of $\sigma_\beta = 0, 0.05, 0.1, 0.15, 0.2$, with $\sigma_\beta = 0$ representing no applied noise. Solid lines are sinusoidal fits to each set of data with fixed periodicity. **c** shows the phases and uncertainties obtained from sinusoidal fits to the fringes in **a** and **b**, where error bars correspond to ±1 standard error in the phase obtained from each sinusoidal fit. The vertical dotted line corresponds to the expected phase of zero radians.

fringes shown agrees to within 6 μ$g$, with a weighted mean of $g_0 = 9.79674$ m s$^{-2}$. More generally, the error-robust pulses reduce the measurement uncertainty by at least a factor of $2 \times$ for all $T$.

Crucially, these data also confirm that our error-robust pulses have the theoretically expected measurement scale factor $\mathcal{S} = 6kT^2$ (neglecting corrections due to the finite duration of the pulses, which we calculate to be of order $(\Omega_{max}T)^{-1}$ using the sensitivity function formalism[41,42] and ignoring losses into higher-order momentum states), which relates the interferometer phase shift to the acceleration via $\phi = \mathcal{S}a$. This is shown in Fig. 2c, where we have plotted the experimentally-determined scale factors for each sequence type for interrogation times of 5, 7.5, and 10 ms alongside the theoretically expected scale factor. The scale factors using error-robust pulses agree to within 1% of the theoretical value. All scale factors are obtained by fitting sinusoids with an additional linear slope to the fringes in Fig. 2a, b and extracting the frequency of the fitted curves.

Although the location of the central fringe − and hence the measured value of gravity − is independent of the scale factor in the chirp-domain measurements shown in Fig. 2, the scale factor sets the frequency of the fringes and hence the sensitivity with which one can determine $g^3$. Crucially, the scale factor obtained using error-robust pulses has the same $T$-squared dependence as conventional pulses, and moreover they enable a more precise determination of the central fringe location by enhancing fringe visibility. Additionally, in dynamic environments one may not have sufficient time to scan a fringe and determine the central fringe location before the acceleration changes. In such cases, knowledge of the proportionality between the interferometer phase and acceleration is critical. More generally, precise knowledge of the scale factor is needed to convert measured phases into inertial signals[7,43,44] and is essential when compensating for the effect of platform vibrations via feedforward or post-correction[6,45].

In aggregate these data demonstrate that tailored pulses obtained from robust control can improve the inertial measurement sensitivity in a cold-atom interferometer, even in near-optimal operating settings. Accordingly, these data rebut a common criticism to the use of robust control in quantum devices − that it may deliver enhancement in some circumstances but in general requires a sacrifice of baseline performance. We have now shown this trade-off is not substantiated when suitably designed control solutions can suppress both intrinsic and extrinsic sources of noise.

## Mitigation of laser-intensity fluctuations

We compare the robustness of each sequence type by experimentally emulating the quasi-static laser intensity variations that such an interferometer could be subjected to in the field. As illustrated in Fig. 1f, constant lateral accelerations acting during the interferometer manifest as a temporal variation in the laser intensity and hence two-photon Rabi frequency experienced by the atoms. The magnitude of constant lateral acceleration required to move an atom initially at rest in the center of a Gaussian beam with a $1/e^2$ radius of $w$ to a region with 20% lower intensity at the time of the final interferometer pulse is $\approx 0.17w/T^2$. For context, this is ~1 $g$ for our beam radius ($w = 5$ mm) and maximum interrogation time ($T = 10$ ms). Transverse accelerations of this magnitude can occur under relatively benign conditions; for example, any strapdown atom interferometer with three orthogonal measurement axes will experience at least 1 $g$ transverse acceleration if one of the beams is oriented at 90˚ to local gravity. They can also be induced by platform accelerations in onboard applications, for example due to the sudden turbulence or banking of an aircraft, or during the motion of a marine vessel in moderate to rough seas (e.g. 5 on the Beaufort scale).

To this end, quasi-static noise of varying intensity is applied to the set-point value of a power servo that stabilizes the intensity of the interferometry laser at the input of the acousto-optic modulator (AOM) used for pulse shaping. Prior to each interferometry pulse, the set-point is varied by multiplying the nominal value of the intensity by a scaling factor $\beta$ selected from a normal distribution with unity mean and a standard deviation $\sigma_\beta$.

The results are shown in Fig. 3 for a $T = 5$ ms order-3 ($6\hbar k$) interferometer composed of either error-robust or Gaussian pulses. The applied noise significantly deteriorates the output of the interferometer employing Gaussian pulses to the point where, at $\sigma_\beta = 0.2$, a sinusoidal fringe is barely resolvable, with a fractional uncertainty in fitted fringe amplitude of 27%. Laser intensity noise of this kind reduces both the precision and accuracy of the phase measurements performed with Gaussian pulses. In comparison, the fringes obtained using error-robust pulses (Fig. 3b) are less impacted by this noise. This is immediately evident in both qualitative examination of the fringes themselves, which show minimal degradation, and in the imputed phases and uncertainties shown in Fig. 3c.

The use of error-robust pulses enables consistently accurate measurements (within 1$\sigma$) of the interferometer phase − which we

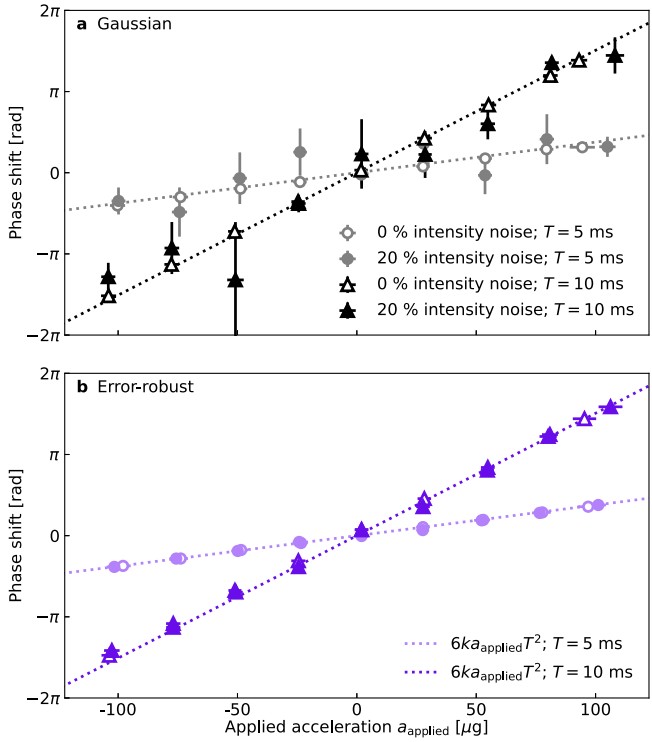

**Fig. 4 | Measurements of applied platform acceleration in the presence of laser intensity noise.** Measured phase shifts obtained from sinusoidal fits to interference fringes for (**a**) Gaussian and (**b**) error-robust pulses as the applied platform acceleration is varied from −100 to +100 μg. The measurements were repeated for interferometers with interrogation times $T = 5$ ms and $T = 10$ ms and for 0 and 20% applied laser intensity noise. Phase shifts are obtained by performing sinusoidal fits with fixed period to interference fringes comprising 33 data points, obtained by varying the DC phase offset of the final interferometry pulse in the interval $[0, 2\pi)$; the value of the applied acceleration $a_{\text{applied}}$ is taken as the average acceleration measured by the classical accelerometer at 40 kSa/s during all the interferometric measurements of duration $2T$ that comprise a fringe, with horizontal error bars of magnitude 3 μg drawn for ±1 standard deviation of these samples. Vertical error bars denote ±1 standard error in the phases obtained from sinusoidal fits to each fringe and are of order 10 μrad with no applied noise, growing to order 100 μrad and 1 rad in the presence of applied noise when using error-robust and Gaussian pulses, respectively. In the absence of intensity noise, we observe a negligible change in fringe visibility as a function of applied acceleration (the 1σ variation in fringe visibility is less than 0.0012 for all sequences and interrogation times). The dotted lines plot $\phi = 6k a_{\text{applied}} T^2$, corresponding to the theoretically expected scale factor of $\mathcal{S} = 6kT^2$.

expect to be zero in the absence of an applied or time-varying acceleration − for all applied noise strengths. In contrast, interferometry performed using Gaussian pulses leads to substantial divergence between the actual and measured phase leading to inaccuracy relative to the 1σ threshold for just 5% laser intensity noise. Stated differently, the interferometer using Gaussian pulses fails to make accurate phase measurements in the presence of such noise.

Furthermore, the error-robust pulses enable a consistently higher measurement precision (the horizontal error bars in Fig. 3c) with and without noise, with factors of improvement relative to their Gaussian counterparts ranging from 2.7 × in the absence of noise and up to 10× for $\sigma_\beta = 0.15$. When Gaussian pulses are employed, the single-shot phase uncertainty (determined by multiplying the phase uncertainty of the sinusoidal fits in Fig. 3a by the square-root of the number of points fitted) grows by an order of magnitude as the level of applied intensity noise is increased, exceeding 1 rad by $\sigma_\beta = 0.1$ (10%), while error-robust pulses the single-shot phase uncertainty increases only to 281 mrad for $\sigma_\beta = 0.2$.

## Robust measurement of applied platform acceleration

Finally, we investigate the improvement offered by error-robust Bragg pulses in measuring deliberately applied accelerations of the mirror that retro-reflects our interferometry laser and serves as the inertial reference in our apparatus. The mirror sits atop a Minus K 50BM-10 vibration isolation stage that is retro-fitted with voice coils to actively drive the stage platform. A classical accelerometer (Silicon Audio 203-120) mounted on the platform provides feedback to the voice coils for active vibration cancellation[46,47]. For the experiments described here, we disabled this feedback and instead drove the voice coils to induce small accelerations in the range of ±100 μg, which remained approximately constant over the interferometer's duration, before reversing the direction of applied acceleration to return the stage to its initial position. Any contribution to the interferometric phase shift due to temporal variations about the mean applied acceleration $a_{\text{applied}}$ is calculated from the classical accelerometer data (neglecting corrections for finite-duration pulses) and accounted for when fitting to interferometric fringes, allowing us to extract the phase shift associated with $a_{\text{applied}}$. We continuously measure the resulting mean platform acceleration $a_{\text{applied}}$ with the classical accelerometer and compare it to the quantum interferometric measurements of the platform acceleration under both normal operating conditions and in the presence of pulse-to-pulse laser intensity fluctuations. By considering vertical accelerations in the ±100 μg range, our interferometric measurements remain within one fringe of the interferometer for interrogation times of relevance to mobile interferometers (~10 ms). Remaining within one fringe means we can avoid fringe ambiguity without, for example, requiring sensor fusion with a classical co-sensor[48] or implementing schemes to extend the dynamic range[49].

Figure 4a, b present the results of these measurements for both conventional Gaussian and error-robust pulse sequences. For each pulse type, we study the interferometric phase shift measured in response to applied accelerations $a_{\text{applied}}$ of varying magnitudes for interferometer interrogation times of $T = 5$ ms and $T = 10$ ms. We derive the interferometer scale factor from the gradient of a linear fit to these data, which agrees with the theoretically expected value $\mathcal{S} = 6kT^2$ to within 2%. Our error-robust pulses maintain this agreement in the presence of large laser intensity fluctuations (shown here for 20% relative laser intensity noise), confirming that the stability of our scale factor is maintained under error-robust pulse sequences. In contrast, for Gaussian pulse sequences the uncertainty in the measured scale factor increases to 24% and 6% for $T = 5$ ms and $T = 10$ ms, respectively, in the presence of 20% laser intensity noise − a significant degradation compared to interferometry using our error-robust pulses.

Regarding precision, measurements using our error-robust pulses generally outperform those using Gaussian pulses. Precision is defined here as the uncertainty in the phase of our fitted fringes, which is represented by the vertical error bars in Fig. 4. In the absence of applied laser intensity noise, our error-robust pulses provide measurements of applied acceleration that are up to 4.9× more precise than those from Gaussian pulses, with average improvements of 3.9× and 2.0× for $T = 5$ ms and $T = 10$ ms, respectively. Under 20% pulse-to-pulse laser intensity variation, our error-robust pulse sequences offer up to 21× improvement in measurement precision over Gaussian pulse sequences, with average improvements of 13× and 7× for $T = 5$ ms and $T = 10$ ms, respectively. These data demonstrate that, unlike Gaussian pulse sequences, our error-robust pulses can deliver highly precise measurements across a range of accelerations even under the large intensity fluctuations that may occur during mobile operation.

## Discussion

We have experimentally demonstrated that tailored error-robust optical pulses can improve the inertial sensitivity of a state-of-the-art atom interferometric sensor operating within a noisy environment.

Crucially, we validated this with two distinct metrological applications: a measurement of Earth's gravity and measurements of applied platform accelerations. In both these cases, we confirm that for error-robust pulses the measurement scale factor closely matches the theoretically predicted value, and that using these pulses reduces measurement uncertainty by up to 21× compared to conventional atom interferometers composed of Gaussian pulses. Furthermore, interferometric phase measurements with error-robust pulses agree with equivalent measurements made using Gaussian pulses to within a 2-$\sigma$ uncertainty window in all cases, putting bounds on any potential systematic bias introduced.

Although open questions remain to be addressed in future experiments, these results confirm that appropriately constructed robust control techniques can suppress two major noise sources encountered in fielded atom interferometric sensors: variation in laser intensity caused by atomic cloud expansion and unwanted platform motion, and inhomogeneous atomic velocity caused by the finite momentum width of the atom source. We also expect that the enhanced velocity acceptance of error-robust pulses would mitigate contrast loss caused by longitudinal platform accelerations in onboard applications, where these accelerations can be large enough to produce appreciable pulse-to-pulse Doppler shifts[12]. The upper bound on the size of longitudinal accelerations that can be mitigated depends upon the interrogation time and pulse velocity acceptance. Consequently, if the acceleration-induced Doppler shift between pulses is larger than the sequence's velocity acceptance, it is likely that active compensation techniques will be required[6,40] to fully mitigate contrast loss.

This work establishes a clear methodology for designing and verifying error-robust control solutions for atom interferometers, and opens a novel pathway towards ruggedized cold-atom sensors for deployment in real-world environments. Our validated approach to error suppression, which exploits open-loop controls designed and executed in software, is extremely flexible and highly configurable, suggesting it is a promising route towards mitigating (or entirely eliminating) other sources of error that are preventing the commercial adoption of mobile cold-atom inertial sensing.

Future work will establish whether any systematic biases are introduced by error-robust pulses below the 6 $\mu g$ threshold set by this work, quantify variations in the measurement scale factor below the 1% level shown in Fig. 2c, and extend our application of error-robust control to interferometers with larger momentum splittings. Reaching larger momentum orders will likely require higher laser powers, tailored concatenated pulse schemes[50], and implementing AC Stark shift mitigation strategies[51]. AC Stark shift mitigation should also improve the performance of our error-robust control sequences, especially at longer interrogation times.

## Methods

### Experimental methods

Atoms are released from a magneto-optical trap (MOT) and undergo polarization-gradient cooling[52] in a nulled magnetic field, resulting in ~$10^9$ atoms at a temperature of ~3 $\mu$K before a 200 mG quantization field is applied along the vertical axis. Atoms are pumped into the $F = 1, m_F = 0$ Zeeman state with $\pi$-polarized light resonant with the $F = 1 \rightarrow F' = 1$ transition and atoms remaining in $F = 2$ are blown away with a brief pulse of resonant MOT light. The interferometry beams perform two concatenated Bloch oscillations that both select a momentum distribution with a (2 × standard deviation) width ~1.6 $\hbar k$ and separate it from the broader source in momentum space, for use as the initial state for interferometry. For readout, we use frequency-modulated imaging[53]; after applying interferometry pulses, the output momentum states are further accelerated by 30 $\hbar k$ in opposite directions by velocity-selective Bloch oscillations[54] so that they are spatially separated when the atoms fall through a light sheet which is phase modulated at 44 MHz with one sideband near-resonant with the cycling $F = 2 \rightarrow F' = 3$ transition. A separate repump beam keeps atoms out of dark states during readout and, upon 44 MHz demodulation of the signal from a fast photodiode onto which the light sheet is focused, a time series with peaks for different momentum states is attained from which relative populations can be inferred.

We realize Bragg pulse sequences with arbitrary amplitude, frequency and phase profiles with an agile RF and laser system (shown in Fig. 5) capable of creating the necessary tailored light pulses with high fidelity. Indeed, this is the only hardware-level innovation[55,56] required in order to realize this form of sensing, as all other operational advances described here are achieved in software. Specifically, time-domain laser pulses are produced by driving a single AOM with a

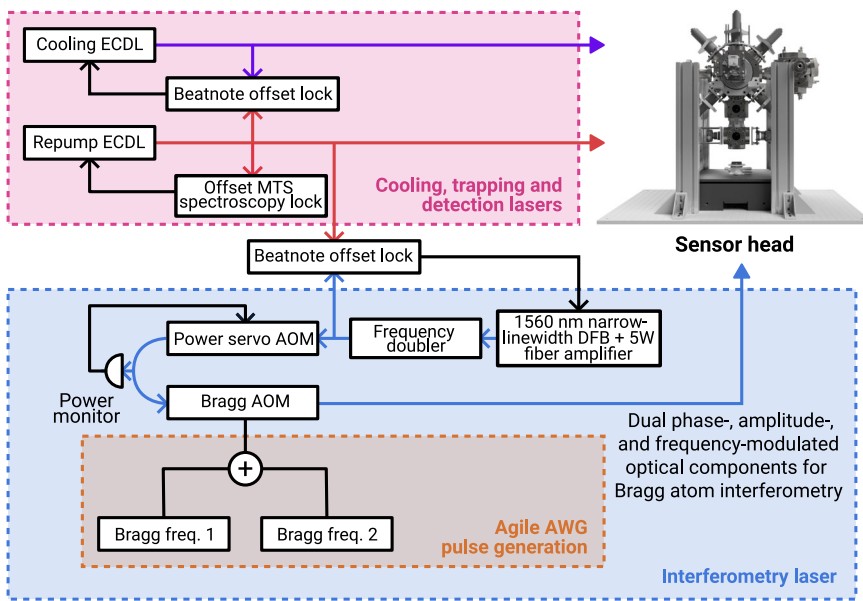

**Fig. 5 | Experimental apparatus.** Diagram of laser system used for atom-interferometric sensing with error-robust Bragg pulses. The upper panel depicts the laser subsystem used for cooling, trapping, and detection of the atomic source of $^{87}$Rb used in this work. The lower panel shows the subsystem used to produce Bragg interferometry sequences with the arbitrary control of phase, amplitude, and frequency needed to realize sequences of error-robust pulses.

numerically synthesized waveform that is reproduced physically on a Tabor Proteus P2584D arbitrary waveform generator (AWG). This waveform is generated from the sum of two sinusoidal frequency components – whose frequency difference is swept to compensate the Doppler shift from gravitational acceleration – with amplitude, phase and frequency modulation applied to each component prior to taking their numerical sum in order to realize arbitrary pulse sequences.

Our use of a single AOM reduces the phase noise that would otherwise accrue between the Bragg lattice components due to separated optical paths if the frequency components were synthesized separately and applied to discrete AOMs. A power servo stabilizes the optical power seeding this AOM, and its output is then fiber-coupled for delivery to the interferometry chamber where the light enters free space as a collimated Gaussian beam with a $1/e^2$ beam diameter of 9.6 mm. The output power at the chamber is dependent on the separation between the Bragg driving frequencies in addition to their amplitude due to variations in AOM diffraction and fiber-coupling efficiencies; we account for this with a look-up table that appropriately scales the amplitudes of the synthesized pulse waveforms to produce the desired output. Although all the interferometry light shares a common polarization in this configuration, upon retro-reflection only a single Bragg lattice remains resonant in the Doppler-shifted atomic rest frame, with all other transitions being off-resonant following ~20 ms of freefall.

We commenced our interferometry sequences 120 ms after release from the MOT to avoid interrogating a region with a significant magnetic gradient in our chamber. Practically, this means we can only reach interrogation times on the order of 10 ms while allowing momentum classes to separate appreciably before state detection.

Under ideal operating conditions with order-1 pulses, our interferometer is able to integrate down to n$g$ precision. Employing Bragg pulses gives the interferometer robustness to quadratic Zeeman shifts[43], allowing the device to operate in a far-from-ideal magnetic environment that exhibits RMS magnetic field fluctuations on the order of 100 μG and static gradients ~0.5 G cm$^{-1}$ without the need for shielding.

## Design of error-robust Bragg pulses

Developing noise-resilient Bragg pulses introduces several specific optimization challenges not present in error-robust control for Raman pulses[27], rendering this problem difficult for most computational methods. Firstly, since Bragg diffraction is an inherently multi-state process, multiple additional levels need to be accounted for in the optimization space; accordingly, well-established techniques for optimal control or composite pulsing derived from NMR are not easily applicable in the Bragg regime. Additionally, the need to add stringent band-limits on candidate solutions in order to ensure controls do not induce population leakage out of the target $|p\rangle$ and $|p+2n\hbar k\rangle$ momentum modes to higher-order states adds a further computational challenge, as this becomes a *constrained optimization* problem beyond the reach of many commonly available optimization engines.

We use Q-CTRL's infrastructure software[57] and its model-based robust control functionality for all Bragg pulse optimization conducted in this work. Complex control design problems are represented using computational data-flow graphs which capture the dependence of a cost function (the quantity we wish to minimize e.g. pulse infidelity) on accessible optimization variables (the control pulse waveform). The control variables, noise channels, constraints, and cost function are all represented as nodes in the graph. We delineate the optimization steps in generating an individual robust Bragg pulse:

1. Identify time-dependent control parameters. In the problem treated here, these are the two-photon Rabi frequency (both its amplitude and phase) and the two-photon detuning. Robust Bragg pulses are described by piecewise-constant waveforms of amplitude $\Omega_R(t)$, phase $\phi_L(t)$, and frequency $\delta(t)$.

2. Define realistic experimental constraints on controls. We apply physically motivated upper bounds on the peak two-photon Rabi frequency $\Omega_{max}$ and the magnitude of the two-photon detuning $\delta_{max}$ for a given shaped pulse. In order to ensure faithful waveform reproduction in hardware and minimize population leakage we also apply a sinc smoothing filter to the control variables $R(t) \equiv \Omega_R(t)\cos[\phi_L(t)]$, $I(t) \equiv \Omega_R(t)\sin[\phi_L(t)]$, and $\delta(t)$, with a maximum cut-off frequency $\omega_{max}$ (80 kHz and 95 kHz for mirrors and beamsplitters, respectively). To apply the filter to a given piecewise-constant control variable $c(t)$, we compute the integral

$$\int_{-\infty}^{\infty} c(t')\frac{\sin[\omega_{max}(t-t')]}{\pi(t-t')}dt' = \frac{1}{2\pi}\int_{-\omega_{max}}^{\omega_{max}} e^{i\omega t}\tilde{c}(\omega)d\omega, \qquad (2)$$

where $\tilde{c}(\omega)$ is the Fourier transform of $c(t)$. This eliminates all frequency components above $\omega_{max}$ in $c(t)$. After the filter is applied, the filtered control variable is re-discretized into a piecewise-constant function. Furthermore, to avoid sudden jumps in laser intensity at waveform edges, we constrain $\Omega_R(t)$ to zero at the start and end of each pulse.

3. Define noise channels for robustness. We target the following sources of pulse infidelity: variations in initial atomic momentum (parameterized by dimensionless momentum detuning $\delta_p \equiv p/\hbar k$) and variations in the amplitude of the two-photon Rabi frequency (parameterized by amplitude error $\beta$ where the effective two-photon Rabi frequency experienced by a given atom is $\Omega_{eff}(t) \equiv (1+\beta)\Omega_R(t)$ and $\Omega_R(t)$ is the intended two-photon Rabi frequency amplitude). We treat the momentum distribution as a Gaussian with standard deviation $\sigma_p$ and the distribution of amplitude errors $\beta$ as uniform with bounds $\beta_{min}$ and $\beta_{max}$. Each noise distribution has a different time-dependence; the momentum distribution is constant for all pulses in the interferometer whereas the amplitude-error distribution is constant only during an individual pulse and allowed to vary between them due to the expansion of the atomic cloud. For the error-robust beamsplitter and mirror pulses used in this paper, $\sigma_p = 0.15\hbar k$ and $\beta_{min}, \beta_{max} = \{-0.15, 0.15\}$.

4. Define and evaluate cost function specific to each pulse. We first define a cost for an arbitrary control applied to an individual atom, and then average this measure over a sample from our noise distributions to obtain an ensemble-average cost. This gives us a measure of how robust a given pulse is to our specific noise distributions.

5. Find control solution (shaped pulse) that minimizes ensemble-average cost node, subject to constraints on candidate solutions. We execute control optimization using an Adam stochastic gradient-descent optimization method[58]. Starting with a guess for our controls (*i.e.* a random initial seed), at each iteration of the Adam method we re-sample our noise distributions to calculate the ensemble-average cost and compute a gradient of infidelity with respect to accessible control variables.

Although no constraint on pulse symmetry is applied during our optimization, we find that many optimized pulses have symmetric or almost symmetric waveforms. A temporally symmetric pulse has a symmetric response in frequency space: the target state excitation probability is identical for an atom which is equally positively or negatively detuned from the resonant frequency. Since we optimize pulses for robustness against symmetric momentum distributions (and hence symmetric two-photon detuning distributions), we speculate that the optimizer finds symmetric solutions because they naturally satisfy this robustness criterion. Similar symmetries are also observed in error-robust pulse design for NMR applications[59].

The choice of cost functions used to optimize robust mirror and beamsplitter pulses as described in step four above represents a critical aspect of the application of robust control to atom interferometry and we therefore expand our discussion of this step. The cost definition for the mirror pulse $\Phi_M$ represents the distance between the optimized pulse propagator $\hat{U}_M$ and the target unitary $\hat{U}_\pi$:

$$\Phi_M \equiv 1 - \left| \frac{\mathrm{Tr}(\hat{U}_\pi^\dagger \hat{U}_M)}{\mathrm{Tr}(\hat{U}_\pi^\dagger \hat{U}_\pi)} \right|^2. \tag{3}$$

In the Bloch-band basis of momentum states $|p + 2n\hbar k\rangle$, the target unitary $\hat{U}_\pi$ for an ideal order-$n$ mirror has matrix elements given by

$$U_{\pi,lm} = \begin{cases} 1, & l = m \\ -i, & l = 0, m = n; l = n, m = 0 \\ 0, & \text{otherwise}. \end{cases} \tag{4}$$

Before evaluating Eq. (3), both $\hat{U}_M$ and $\hat{U}_\pi$ are pre-multiplied by a projection operator $\hat{P}$ defined through its matrix elements:

$$P_{lm} = \begin{cases} 1, & l = m = 0, n \\ 1, & l = 0, m = n; l = n, m = 0 \\ 0, & \text{otherwise}. \end{cases} \tag{5}$$

This projects both operators onto the relevant momentum subspace and means that our mirror cost is only sensitive to transformations that affect the momentum states $|p\rangle$ and $|p + 2n\hbar k\rangle$ that form the two interferometer arms.

In contrast, the cost definition for the beamsplitter pulses requires special consideration of pulse phase in order to deliver the appropriate action of the pulse pair. This can be understood as follows. Given atoms initially in $|p\rangle$, the first beamsplitter must produce an equal superposition of two momentum states $|p\rangle$ and $|p + 2n\hbar k\rangle$ with relative phase $\phi_1(\delta_p, \beta)$. The final beamsplitter must transform a superposition of two momentum states to a final state $|\psi_f\rangle$ where the probabilities $P_{1,2}$ to be found in momentum states $|p\rangle$ and $|p + 2n\hbar k\rangle$, respectively, satisfy:

$$P_1 - P_2 = \cos(\phi_{int} + n\phi_{BS} + \phi_2 - \phi_1). \tag{6}$$

$\phi_{1,2}(\delta_p, \beta)$ are superposition phases which in general depend on the initial momentum detuning $\delta_p$ and two-photon Rabi frequency

amplitude error $\beta$, $\phi_{BS}$ is a DC offset in the laser phase during the pulse, and $\phi_{int}$ is the acceleration-dependent interferometer phase. For arbitrary beamsplitter pulses, $(\phi_2 - \phi_1)$ may not cancel, yielding an interferometer fringe that varies as a function of atomic momentum (a case unlike the use of identical Gaussian pulses) and which would result in a washing out of the fringe following an ensemble average. We thus account for this by simulating the entire sequence, assuming a perfect mirror pulse, and introducing a variable DC phase offset $\phi_{BS}/2$ in the mirror phase to obtain a fringe given by Eq. (6). This fringe is averaged over the two noise processes and the final ensemble-average cost is taken as 1- the dot-product between the ensemble-averaged fringe and the target fringe with ideal visibility, phase, and offset.

Intra-pulse phase variations can in principle introduce spurious phase shifts to the interference fringes. If the atoms do not see the exact waveform (phase, intensity, and frequency profile) obtained from the optimization, then the fidelity of the pulse is lowered, introducing systematic phase shifts into the measurement.

We quantified this in numerical simulation by applying Gaussian phase noise to each (1 μs) piecewise-constant time segment of our light pulse sequence and calculating the standard deviation of the resulting interferometer phase. We found that Gaussian phase noise with $1\sigma = 0.1$ mrad results in a maximum interferometer phase error of 0.12 and 0.19 mrad for Gaussian and error-robust pulses, respectively. This is well below the measurement noise on our fringes.

Phase resolution of arbitrary wave synthesis depends on many factors, including phase stability of the oscillator and finite sample rate and amplitude resolution. Many of these result in a synthesized phase error that is negligible compared to the phase uncertainties we quantify here. Nevertheless, as a concrete example for the above result, one could consider a per segment phase error of 0.1 mrad (e.g. a gross upper bound of the effect of 16-bit amplitude digitization of our AWG).

## Numerical model of individual Bragg pulses

Figure 6 depicts an energy-momentum diagram for an order-$n$ Bragg transition, where atomic states which differ in momentum by $2n\hbar k$ are coupled via simultaneous absorption and emission of $n$ photons from two-counter propagating laser beams with frequencies $\omega_{1,2}$ and average wavenumber $k = \frac{1}{2c}(\omega_1 + \omega_2)$. The dynamics of a Bragg transition may be conveniently represented in the Bloch-band basis of atomic momentum states $|p + 2m\hbar k\rangle$, where $m \in \mathbb{Z}$ and $p$ is the initial atomic momentum along the beam axis[37]. In this basis, the Hamiltonian $\hat{H}(t)$

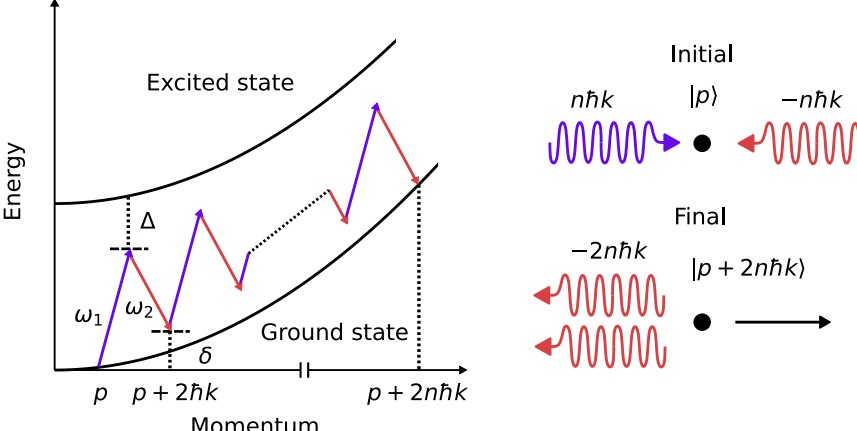

**Fig. 6 | Schematic of an order-$n$ multi-photon Bragg transition.** (Left) Energy-momentum diagram for an order-$n$ Bragg transition. Two counter-propagating beams with frequencies $\omega_{1,2}$ are detuned by an amount $\Delta$ from an upper excited state. If the laser frequency difference $\delta \equiv \omega_1 - \omega_2$ satisfies $\delta = 4n\omega_R + 2pk/M$, $n$

photons are resonantly absorbed from one beam and emitted into the other, providing a net momentum kick of $2n\hbar k$ to an atom initially with momentum $p$, while leaving its electronic state unchanged (right).

describing the atom-light interaction is tridiagonal with non-zero matrix elements given by[35,37,60]

$$H_{mn}(t) = \begin{cases} \Omega_R(t)e^{i\phi_L(t)/2}, & m = n-1 \\ \delta_m(t), & m = n \\ \Omega_R(t)e^{-i\phi_L(t)/2}, & m = n+1. \end{cases} \quad (7)$$

We have defined the generalized detuning $\delta_m(t) \equiv \omega_R(2m + \delta_p + \delta(t)/4\omega_R)^2$. $\delta_p \equiv p/\hbar k$ is a dimensionless detuning due to the initial atomic momentum and $\delta(t)$ is the time-dependent two-photon detuning (laser frequency difference). $\omega_R \equiv \hbar k^2/(2M)$ is the single-photon recoil frequency and $M$ is the atomic mass. $\Omega_R$ and $\phi_L$ are the amplitudes and phases of the complex two-photon Rabi frequency that defines the coupling between adjacent momentum states. Note that there is a one-to-one mapping between the momentum basis and Bloch-band basis via $p = \hbar k(m + \delta_p)$.

Since we consider only piecewise-constant Bragg pulses in this work, the operation of individual Bragg pulses and entire interferometer sequences on arbitrary momentum states may be simulated straightforwardly as described in ref. [26]. The quantum state $|\psi(t + \Delta t)\rangle$ following evolution under a piecewise-constant Hamiltonian $\hat{H}(t)$ for a time-step with duration $\Delta t$ is given by $|\psi(t + \Delta t)\rangle = \hat{U}(t)|\psi(t)\rangle$, where the propagator is defined by the matrix exponential $\hat{U}(t) \equiv \exp(-i\hat{H}(t)\Delta t)$. The transformation due to an entire pulse of duration $t_f - t_0$ may then be represented by a single combined propagator given by the product of propagators calculated for each time-step in the pulse. Explicitly, $\hat{U}(t_f, t_0) = \hat{U}(t_f - \Delta t)\hat{U}(t_f - 2\Delta t) \ldots \hat{U}(t_0 + \Delta t)\hat{U}(t_0)$.

The dynamics of Eq. (7) are not closed, and for the $n = 3$ transitions simulated in this work we therefore truncate our basis such that both extremal momentum states are negligibly populated[38,61].

## Data availability
The processed data used to generate Figs. 2, 3, and 4 are provided in the Source Data file. The raw data and analysis code are available from the corresponding author upon request. Source data are provided with this paper.

## Code availability
The code used to reproduce figures in this study is available from the corresponding author upon request. The error-robust light pulses in this study were developed using Q-CTRL's proprietary quantum infrastructure software which is available upon purchase of a license.

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

## Acknowledgements

The authors are grateful to all other colleagues at Q-CTRL whose technical, product engineering, and design work has supported the results presented in this paper.

## Author contributions

All authors (i.e. J.C.S., M.S.C., P.S.L., S.S.S., A.R.M., K.S.G., M.L.G., V.S.P., N.M.W., C.D.M., A.R., P.J.E., N.P.R., R.P.A., M.R.H., and M.J.B.) conceived and designed the experiments (including developing the experimental apparatus). J.C.S., M.S.C., P.S.L., and N.P.R. performed the experiments. J.C.S., M.S.C., P.S.L., S.S.S., K.S.G., N.P.R., R.P.A., M.R.H., and M.J.B. analyzed and/or critically evaluated the data. J.C.S., M.S.C., P.S.L., S.S.S., N.P.R., R.P.A., M.R.H., and M.J.B. wrote the manuscript.

## Competing interests

J.C.S., M.S.C., P.S.L., S.S.S., K.S.G., N.M.W., C.D.M., A.R., P.J.E., N.P.R., R.P.A., M.R.H., and M.J.B. are associated and/or hold shares/options with quantum technology company Q-CTRL Pty. Ltd. that will make use of some of the findings of this article in the quantum sensors they develop. The remaining authors declare no other competing interests.
