## [Peer Review File · Nature Communications]

Enhancing the sensitivity of atom-interferometric inertial sensors using robust controlREVIEWER COMMENTS

Reviewer #1 (Remarks to the Author):

The authors present experimental work related to the improvement of atom interferometric sensors using tailored atom-optical pulses. This work expands upon previous theoretical and experimental studies of both composite and optimal control pulses in atom interferometry. The authors aim to address the important question of whether tailored light pulses are useful for inertial measurements: do they enhance the atomic fringe visibility at the expense of other important quantities, such as the scale factor or phase uncertainty? This work takes an important step in that direction by demonstrating that tailored Bragg pulses can, in fact, produce relevant inertial measurements. Additionally, they demonstrate enhanced phase sensitivity compared to conventional light pulses in the presence of simulated environmental noise while maintaining a consistent scale factor.

The manuscript is clear and well written. The results reported are significant and will be important to the quantum sensing community as this quantum technology evolves from laboratory prototypes to marketable products for real-world applications.

I recommend the manuscript for publication provided the authors address the following points.

1) Lines 19-27: The authors should consider citing the following paper:

T. Wilkason et al, Phys. Rev. Lett. 129, 183202 (2022).

2) Line 39: “uncontrolled motion due to platform acceleration”. This statement could be interpreted as platform vibrations that randomize the phase of the atom interferometer, which I do not believe is the intention here. It would be helpful to specify what type of accelerations the authors are referring to. Are they transverse or longitudinal to the direction of the interferometry beam? Are they slow or fast relative to the duration of the interferometer? Are they deterministic or random?

3) Line 96-97: “...error-robust pulses introduce no additional bias to within the

measurement uncertainty.” What is the measurement uncertainty here? $6 \mu\text{g}$? Please consider stating it directly here.

4) Line 100-101: “This manifests as a linear trend over the range of sweep rates measured...”. It’s not clear why one should expect a linear trend on top of the sinusoidal fringe when varying the sweep rate, α . Is this being caused by a variation in the fringe offset or visibility? When $\alpha = 6 \cdot k \cdot g$, at the location of the common dark fringe, the fringe visibility should be maximum. Although this resonance can be quite broad relative to the fringe spacing, the visibility should decrease on either side of the dark fringe provided the peak of the velocity distribution is addressed by the first beam splitter pulse. It’s less intuitive to understand how the fringe offset is affected by α . Could the authors provide further clarification in this paragraph?

5) Figure 2 and caption: It would be more transparent to show the residuals (data – model) in Fig. 2c, since it is not obvious how well the two data sets match the model $6 \cdot k \cdot T^2$. This would also address the fact that the caption refers to error bars that are not visible in this figure.

6) Line 112: “...(neglecting contributions of order $(\Omega_{\text{max}} \cdot T)^{-2}$)...”. What is the value of Ω_{max} for these data? How large are these contributions for your conventional Gaussian pulses? It would be useful to point out that a precise determination of the interferometer scale factor can be determined for any temporal pulse shape using the sensitivity function approach, as described in the following articles:

P. Cheinet et al, IEEE Trans. Instrum. Measure. 57, 1141–1148 (2008).

B. Fang et al, New J. Phys. 20, 023020 (2018).

7) Line 115: “The scale factors using error-robust pulses agree to within 1% of the theoretical value.” How well do the Gaussian pulses agree with the theoretical scale factor? How well do the two sets of measured scale factors agree with one another? This will become clearer after addressing point 5) above.

8) In relation to points 6) and 7) above, the authors should discuss to what extent the scale

factor is important when making inertial measurements. When using a phase-continuous sweep rate, the phase of the interferometer becomes $\Phi = (k_{\text{eff}}g - \alpha)S/k_{\text{eff}}$, where $k_{\text{eff}} = 6k$ and S is the full interferometer scale factor. At the location of the dark fringe, $\Phi = 0$ and one obtains gravity from $g = \alpha/k_{\text{eff}}$, which is independent of S . This is only true in the absence of systematic phase shifts, however.

9) Figure 4: It would be advantageous to show how the fringe visibility varies with applied acceleration for the two cases. The authors should discuss the behaviour of these data on page 6, and consider showing them in Figure 4.

10) Figure 4 caption: "...with horizontal error bars drawn for ± 1 standard deviation...". The horizontal error bars are not visible in the figure, nor are the vertical error bars in Fig. 4b. Please state the typical size of both uncertainties in the caption.

11) Lines 191-198: While this work makes important progress for tailored light pulses, it does not address important metrological questions: do tailored pulses introduce any systematic biases? How does one deal with them? Do they modify the interferometer scale factor beyond the 1% level? In the conclusion, the authors should acknowledge that further studies are needed to answer these questions.

12) Line 235: "In synthesising the pulse waveforms...". This sentence is unclear in the context of the surrounding text. Could the authors please elaborate.

13) Line 241: "Employing Bragg pulses gives the interferometer first-order robustness to magnetic field fluctuations...". This is also the case for Raman pulses addressing $|F, mF = 0\rangle$ to $|F \pm 1, mF = 0\rangle$ transitions between hyperfine ground states. Can the authors elaborate about the intrinsic advantage to using Bragg pulses in the context of rejecting magnetic field effects?

14) Typo on Line 298: missing opening bracket on " $|p + 2n\hbar k\rangle$ ".

15) Typo in caption of Figure 6: "...from an upper excited sate." Should be "state".

Referee comment to manuscript entitled : *“Enhancing the sensitivity of atom-interferometric inertial sensors in dynamic environments using robust control”*

General comment :

In this manuscript, the authors present the use of optimal robust control techniques to enhance the sensitivity of a vertical Mach-Zehnder light-pulse atom interferometer using Bragg transitions as mirror and beam splitter pulses. They claim that **this technique could benefit quantum cold atom inertial sensors dedicated to in-field applications** where performances of these sensors in terms of sensitivity are known to be degraded by several orders of magnitude when going from lab-based to field environment.

It is very interesting for the atom interferometry community to see that applying robust control techniques to a Mach-Zehnder Bragg (order 3) interferometer allows to improve the sensitivity of the interferometer while submitted to noise sources . **This is in my opinion a nice demonstration of the power and utility of robust control techniques** to enhance the performances of cold atom interferometers **especially for Large Momentum Transfer atom optics applications where high fidelity is mandatory.**

Overall, the authors present valuable scientific results **but I do not think it meets requirements for publication in Nature Communications** and I would encourage them to address a more specific/technical journal where their results could be published and address the specific community of atom interferometry.

Additionally, emulating transverse accelerations with intensity variations, is in my opinion far too much a reductive approach of dynamic measurements as this is certainly not the main limitation preventing a cold atom sensor to operate out of the laboratory.

Indeed, the main issue with transverse motion of the atoms in a dynamic environment are the rotations which severely reduces the fringe contrast. Thus, contrary to a lab-based sensor where the benefit of this technique would be immediate, I am very sceptical about its benefits for dynamic operations if rotation issues are not fixed.

Specific remarks :

The manuscript is well written and could be easily resubmitted to another journal without too much to change.

Mitigation of laser intensity fluctuations

- The authors mention “a constant lateral acceleration of magnitude ~ 1 g is sufficient to push an atom initially at rest in the center of the beam”. In the context of onboard application, it would be interesting to know what kind of typical environment give rise to 1g lateral acceleration ?

Robust measurement of applied platform acceleration

- Surprisingly 100 μ g vertical acceleration is very small (in comparison with 1 g transverse acceleration). One would have expected the atom accelerometer to be tested with a much broader acceleration magnitude (extending multiple fringe periods) such as the one

encountered on references [13-14] for example with typical acceleration values linked to a representative case-study in terms of acceleration magnitude and frequency.

Can the authors explain why they chose this acceleration magnitude ?

Methods :

- Atom selection in $mF=0$

It is not so often that laser light is used to pump the atoms in $mF=0$.

Why not using Micro-Wave pulse selection ?

- Only one AOM is used to generate the lattice for Bragg pulses.
If I understand well, then two lattices with opposite velocities are generated?
How do you cope with the parasitic lattice ?

- Optimal pulse sequence

For optimal pulse, the phase is varied by radians inside the pulse. One would expect somehow that this phase enters the final phase shift calculation in the interferometer.

Could you comment on :

- a) the required level of control of the phase
- b) how it enters or not the atom interferometer phase shift ?

Reviewer #3 (Remarks to the Author):

This paper reports on the application of robust quantum control to a Bragg atom interferometer for mobile sensing. Mobile sensing based on atom interferometry is highly promising and has attracted considerable interest. However, operating a mobile atom interferometer introduces significant challenges, as platform motion tends to dramatically degrade the measurement signal. For example, as the authors note, lateral platform motion can cause the atoms to experience temporally varying laser intensity.

The authors numerically design optimized pulses tailored to be robust against relevant noise sources using a suite of optimization tools that their company (Q-CTRL) has developed. They experimentally demonstrate that incorporating these pulses into their atom interferometer offers order of magnitude level improvements in robustness against various types of noise. Critically, they verify that scale factor stability is maintained. This is an excellent achievement. First, in only one other circumstance has robust quantum control been experimentally shown to improve an atom interferometer (ref. 30). Notably, ref. 30 used Raman pulses, which are much simpler than Bragg pulses due to the multilevel structure of the Bragg interaction, for which many momentum states need to be included in the Hamiltonian. As the authors outline, Bragg interferometers offer important advantages, and the experimental demonstration of a benefit from robust control applied to the complex Bragg system is impressive. Second, never before has anyone demonstrated the benefit of robust control to an actual atom interferometry based inertial sensor.

I expect that this paper will have a broad impact. As the authors describe, mobile sensing based on atom interferometry has a wide range of applications of interest to multiple disciplines. The work reported here will likely inspire others working in this field to adopt robust control techniques. The paper is well-written and can be clearly followed by the reader. For the reasons stated above, I believe that this paper would be a valuable contribution to Nature Communications, provided that the questions and comments below are addressed.

-In Fig. 1c, the amplitude of the optimized pulse appears to be approximately, but not

exactly, symmetric in time about the midpoint of the pulse. Could the authors comment on this, and whether there is any reason for this qualitative feature?

-What motivated the particular choice of third order Bragg diffraction, as opposed to higher (or lower) order? Did the authors try other diffraction orders?

-Could the authors comment on what limited T to 10 ms?

-It would be useful if the authors could comment on what hardware calibration, if any, was needed for the laser system to faithfully reproduce the robust pulses.

-In the paragraph beginning on line 175, it appears that the average improvements offered by robust pulses decreases with increasing T . Do the authors have an explanation for this behavior?

-The authors study robustness against laser intensity noise that varies up to 20% from pulse to pulse to model transverse motion. For the beam size the authors use and the time scales involved, this corresponds to robustness against impressively large transverse accelerations. What about robustness against similarly large longitudinal accelerations (which one might also expect in inertial navigation applications), which could lead to Doppler shifts that vary from pulse to pulse? The section on applied platform acceleration is relevant to this point, though for smaller longitudinal accelerations as compared to the transverse accelerations needed for 20% intensity variations from pulse to pulse. Do the authors have any thoughts on whether robustness to longitudinal platform accelerations would be maintained for larger longitudinal accelerations or for longitudinal accelerations that significantly vary in time over the course of the interferometer?

-The authors have demonstrated robustness against the effects of transverse displacements of the platform. Have they considered whether their methods could be applied to improve robustness against time varying platform tilts, which one might also expect to occur in a mobile sensor? This might be more challenging, since tilts change the direction of the momentum kicks received by the atoms, but I would be curious to hear any thoughts the authors might have on this.

-In line 249, the authors state that they added stringent band-limits on candidate solutions to reduce population leakage out of the target momentum states. It would be useful if the authors could discuss this point quantitatively, and in greater detail.

We thank all reviewers for their time and constructive comments, which we address point-by-point below:

Reviewer #1

The authors present experimental work related to the improvement of atom interferometric sensors using tailored atom-optical pulses. This work expands upon previous theoretical and experimental studies of both composite and optimal control pulses in atom interferometry. The authors aim to address the important question of whether tailored light pulses are useful for inertial measurements: do they enhance the atomic fringe visibility at the expense of other important quantities, such as the scale factor or phase uncertainty? This work takes an important step in that direction by demonstrating that tailored Bragg pulses can, in fact, produce relevant inertial measurements. Additionally, they demonstrate enhanced phase sensitivity compared to conventional light pulses in the presence of simulated environmental noise while maintaining a consistent scale factor.

The manuscript is clear and well written. The results reported are significant and will be important to the quantum sensing community as this quantum technology evolves from laboratory prototypes to marketable products for real-world applications.

I recommend the manuscript for publication provided the authors address the following points.

- 1) Lines 19-27: The authors should consider citing the following paper:
T. Wilkason et al, Phys. Rev. Lett. 129, 183202 (2022).

We have added this reference, with the final sentence in the paragraph now reading:
“Notable exceptions include Refs. ~\cite{Wilkason:2022}~and~\cite{Saywell2020b}, which respectively deploy composite Floquet pulses and tailored Raman pulses to improve the pulse fidelity and fringe contrast in Mach-Zehnder interferometers.”

- 2) Line 39: “uncontrolled motion due to platform acceleration”. This statement could be interpreted as platform vibrations that randomize the phase of the atom interferometer, which I do not believe is the intention here. It would be helpful to specify what type of accelerations the authors are referring to. Are they transverse or longitudinal to the direction of the interferometry beam? Are they slow or fast relative to the duration of the interferometer? Are they deterministic or random?

We now explicitly identify “platform accelerations transverse to the propagation direction of the interferometry beams”. Our pulses provide robustness to transverse accelerations irrespective of whether they are slow, fast, deterministic, or random. More detailed discussion is provided in the “MITIGATION OF LASER-INTENSITY FLUCTUATIONS” section.

- 3) Line 96-97: “...error-robust pulses introduce no additional bias to within the measurement uncertainty.” What is the measurement uncertainty here? 6 ug? Please consider stating it directly here.

This has been amended for clarity: “...while the common center indicates that any additional bias introduced by the error-robust pulses is no greater than the $6\ \mu\text{g}$ range of imputed fringe centers for these data.”

4) Line 100-101: “This manifests as a linear trend over the range of sweep rates measured...”. It’s not clear why one should expect a linear trend on top of the sinusoidal fringe when varying the sweep rate, α . Is this being caused by a variation in the fringe offset or visibility? When $\alpha = 6k^*g$, at the location of the common dark fringe, the fringe visibility should be maximum. Although this resonance can be quite broad relative to the fringe spacing, the visibility should decrease on either side on either size of the dark fringe provided the peak of the velocity distribution is addressed by the first beam splitter pulse. It’s less intuitive to understand how the fringe offset is affected by α . Could the authors provide further clarification in this paragraph?

The linear trend discussed in this section is observed in the fringe offset, not the visibility. The same phenomenon can be seen in Müller *et al.* Phys. Rev. Lett. **100**, 180405 (2008). While those authors did not draw attention to this systematic or proffer an explanation, we posit that it may be related to the effect of the Doppler shift on the atomic velocity distribution caused by the mismatch between the chirp rate and the longitudinal acceleration. We have clarified the text, noting that further investigation is required to fully characterize this effect.

“Using error-robust pulses also reduces systematics that are present in these chirp-domain interference fringes. We observe a linear trend in the fringe offset atop the sinusoidal oscillations, highlighted by the shaded regions between the fringes and sinusoidal fits in Fig.~\ref{fig:5}b. If unaccounted for during least-squares regression, this trend shifts the location of the imputed fringe center increasingly as ST is reduced. Using error-robust pulses reduces the slope of the linear trend by at least $2.5\times$ for all values of ST shown here. This phenomenon has been observed in Bragg pulse interferometers~\cite{Muller2008a} and to our knowledge its origin is yet to be identified.

We posit that the linear trend may be related to the pulses being off-resonant when the sweep rate does not perfectly match the Doppler shift caused by longitudinal acceleration. Consequently, the velocity distribution excited by the laser pulses varies with the sweep rate, more so for Gaussian pulses than for error-robust pulses, which are less sensitive to Doppler shifts. Further investigation is required to fully characterize this effect, which could be detrimental when operating in a dynamic environment employing mid-fringe locking techniques~\cite{Templier:Thesis}.”

5) Figure 2 and caption: It would be more transparent to show the residuals (data – model) in Fig. 2c, since it is not obvious how well the two data sets match the model $6k^*T^2$. This would also address the fact that the caption refers to error bars that are not visible in this figure.

Figure 2c has been updated to display the residuals inset.

6) Line 112: "...(neglecting contributions of order $(\Omega_{\text{max}}T)^{-2}$)...". What is the value of Ω_{max} for these data? How large are these contributions for your conventional Gaussian pulses? It would be useful to point out that a precise determination of the interferometer scale factor can be determined for any temporal pulse shape using the sensitivity function approach, as described in the following articles:

P. Cheinet et al, IEEE Trans. Instrum. Measure. 57, 1141–1148 (2008).

B. Fang et al, New J. Phys. 20, 023020 (2018).

There was a typo in our expression for the scale-factor corrections due to pulses with finite duration: $(\Omega_{\text{max}}T)^{-2}$ should read $(\Omega_{\text{max}}T)^{-1}$. We have updated this in the revised manuscript and pointed out that these corrections (which are of order 0.5%, 0.3%, and 0.2% for our Gaussian pulses for $T = 5, 7.5,$ and 10ms respectively) may be calculated using the sensitivity function approach for conventional two-mode interferometers with no multi-path interference effects:

"Crucially, these data also confirm that our error-robust pulses have the theoretically-expected measurement scale factor $\mathcal{S} = 6 k T^2$ (neglecting corrections due to the finite duration of the pulses, which we calculate to be of order $(\Omega_{\text{max}} T)^{-1}$ using the sensitivity function formalism~\cite{Fang:2018,Cheinet2008} and ignoring losses into higher-order momentum states), which relates..."

Note that the standard sensitivity function approach is not obviously applicable here due to the presence of multi-path interference in Bragg diffraction. The degree of multi-path interference, and its effect on the interferometer scale factor, depends on the interrogation time, pulse efficiencies, and the atomic source momentum width. Providing a quantitatively accurate estimate of the scale factor at the level of the 1% deviations shown for Gaussian pulses in the inset to Fig. 2 c would therefore require more detailed knowledge of our cloud momentum width, and the precise degree of multi-path interference, which is unfortunately not accessible from these data.

Finally, we have also updated the first paragraph of this section to clarify the pulse parameters:

"As a first experimental test, we verify the measurement scale factor given by our error-robust Bragg pulses by measuring Earth's gravitational field. We operate the error-robust pulses at a peak two-photon Rabi frequency of $\Omega_{\text{max}} = 2\pi \times 40\text{kHz}$ --- the value for which they were designed --- with no additional calibration, and compare them to order-3 Bragg pulses with a Gaussian profile given by Eq.~\ref{eq: gaussian pulse}, where σ_{τ} is fixed at $25\text{ }\mu\text{s}$ and the amplitude of each pulse is varied to tune the pulse area so as to maximize contrast. Figure~\ref{fig:5}a-b, shows interference fringes for both the error-robust and Gaussian order-3 Bragg pulses..."

7) Line 115: “The scale factors using error-robust pulses agree to within 1% of the theoretical value.” How well do the Gaussian pulses agree with the theoretical scale factor? How well do the two sets of measured scale factors agree with one another? This will become clearer after addressing point 5) above.

As the reviewer suggests, this is now made apparent by the addition of residuals to Fig. 2c. The measured scale factors for Gaussian pulses agree with the theoretical scale factor (in the instantaneous-pulse limit) to within 1.5%, and the two sets of scale factors agree with one another to within 2%.

8) In relation to points 6) and 7) above, the authors should discuss to what extent the scale factor is important when making inertial measurements. When using a phase-continuous sweep rate, the phase of the interferometer becomes $\Phi = (k_{\text{eff}}g - \alpha)S/k_{\text{eff}}$, where $k_{\text{eff}} = 6k$ and S is the full interferometer scale factor. At the location of the dark fringe, $\Phi = 0$ and one obtains gravity from $g = \alpha/k_{\text{eff}}$, which is independent of S . This is only true in the absence of systematic phase shifts, however.

We have included the following discussion on the role of the scale factor in inertial measurements:

“Although the location of the central fringe --- and hence the measured value of gravity --- is independent of the scale factor in the chirp-domain measurements shown in Figure \ref{fig:5}, the scale factor sets the frequency of the fringes and hence the sensitivity with which one can determine g ~\cite{Menoret:2018}. Crucially, the scale factor obtained using error-robust pulses has the same ST^2 -squared dependence as conventional pulses, and moreover they enable a more precise determination of the central fringe location by enhancing fringe visibility. Additionally, in dynamic environments one may not have sufficient time to scan a fringe and determine the central fringe location before the acceleration changes. In such cases, knowledge of the proportionality between the interferometer phase and acceleration is critical. More generally, precise knowledge of the scale factor is needed to convert measured phases into inertial signals~\cite{Gustavson1997, Altin2013, Sorrentino:2014} and is essential when compensating for the effect of platform vibrations via feedforward or post-correction~\cite{Lautier2014, Templier:2022}.”

9) Figure 4: It would be advantageous to show how the fringe visibility varies with applied acceleration for the two cases. The authors should discuss the behaviour of these data on page 6, and consider showing them in Figure 4.

In the absence of intensity noise, we observe a negligible change in fringe visibility as a function of applied acceleration (the 1σ variation in fringe visibility is less than 0.0012 for all sequences and interrogation times). This is also evidenced by the vertical error bars in Figure 4, which we have now described in the caption (see response to point 10 directly below).

10) Figure 4 caption: "...with horizontal error bars drawn for +/-1 standard deviation...". The horizontal error bars are not visible in the figure, nor are the vertical error bars in Fig. 4b. Please state the typical size of both uncertainties in the caption.

The magnitude of the error bars has been added to the caption, as suggested:

"...with horizontal error bars of magnitude $\SI{3}{\micro \textit{g}}$ drawn for ± 1 standard deviation of these samples. Vertical error bars denote ± 1 standard error in the phases obtained from sinusoidal fits to each fringe and are of order $\SI{10}{\micro \textit{rad}}$ with no applied noise, growing to order $\SI{100}{\micro \textit{rad}}$ and $\SI{1}{\textit{rad}}$ in the presence of applied noise when using error-robust and Gaussian pulses, respectively. In the absence of intensity noise, we observe a negligible change in fringe visibility as a function of applied acceleration (the 1σ variation in fringe visibility is less than 0.0012 for all sequences and interrogation times)."

11) Lines 191-198: While this work makes important progress for tailored light pulses, it does not address important metrological questions: do tailored pulses introduce any systematic biases? How does one deal with them? Do they modify the interferometer scale factor beyond the 1% level? In the conclusion, the authors should acknowledge that further studies are needed to answer these questions.

We have updated the manuscript to explicitly state how this work puts bounds on any systematic biases introduced by these pulses and have acknowledged that outstanding questions remain to be addressed in future work:

"...using these pulses reduces measurement uncertainty by up to $21 \times$ compared to conventional atom interferometers composed of Gaussian pulses. Furthermore, interferometric phase measurements with error-robust pulses agree with equivalent measurements made using Gaussian pulses to within a 2σ uncertainty window in all cases, putting bounds on any potential systematic bias introduced."

"Future work will establish whether any systematic biases are introduced by error-robust pulses below the $\SI{6}{\micro \textit{g}}$ threshold set by this work, quantify variations in the measurement scale factor below the 1% level shown in Figure~\ref{fig:5}c, and extend our application of error-robust control to interferometers with larger momentum splittings."

12) Line 235: "In synthesising the pulse waveforms...". This sentence is unclear in the context of the surrounding text. Could the authors please elaborate.

The sentence has been updated:

“The output power at the chamber is dependent on the separation between the Bragg driving frequencies in addition to their amplitude due to variations in AOM diffraction and fibre-coupling efficiencies; we account for this with a look-up table that appropriately scales the amplitudes of the synthesized pulse waveforms to produce the desired output.”

13) Line 241: “Employing Bragg pulses gives the interferometer first-order robustness to magnetic field fluctuations...”. This is also the case for Raman pulses addressing $|F, mF = 0\rangle$ to $|F \pm 1, mF = 0\rangle$ transitions between hyperfine ground states. Can the authors elaborate about the intrinsic advantage to using Bragg pulses in the context of rejecting magnetic field effects?

This was an error, we meant to discuss quadratic Zeeman shifts. We have updated the text to *“Employing Bragg pulses gives the interferometer robustness to quadratic Zeeman shifts~\cite{Altin2013},”*

14) Typo on Line 298: missing opening bracket on “ $|\rho + 2*n*\hbar*k\rangle$ ”.

Corrected.

15) Typo in caption of Figure 6: “...from an upper excited sate.” Should be “state”.

Corrected.

Reviewer #2

(report attached as pdf)

General comment:

In this manuscript, the authors present the use of optimal robust control techniques to enhance the sensitivity of a vertical Mach-Zehnder light-pulse atom interferometer using Bragg transitions as mirror and beam splitter pulses. They claim that this technique could benefit quantum cold atom inertial sensors dedicated to in-field applications where performances of these sensors in terms of sensitivity are known to be degraded by several orders of magnitude when going from lab-based to field environment.

It is very interesting for the atom interferometry community to see that applying robust control techniques to a Mach-Zehnder Bragg (order 3) interferometer allows to improve the sensitivity of the interferometer while submitted to noise sources. This is in my opinion a nice demonstration of the power and utility of robust control techniques to enhance the performances of cold atom interferometers especially for Large Momentum Transfer atom optics applications where high fidelity is mandatory.

Overall, the authors present valuable scientific results but I do not think it meets requirements for publication in Nature Communications and I would encourage them to address a more specific/technical journal where their results could be published and address the specific community of atom interferometry.

Additionally, emulating transverse accelerations with intensity variations, is in my opinion far too much a reductive approach of dynamic measurements as this is certainly not the main limitation preventing a cold atom sensor to operate out of the laboratory.

Indeed, the main issue with transverse motion of the atoms in a dynamic environment are the rotations which severely reduces the fringe contrast. Thus, contrary to a lab-based sensor where the benefit of this technique would be immediate, I am very sceptical about its benefits for dynamic operations if rotation issues are not fixed.

We agree that rotations are a significant challenge for mobile cold-atom sensing onboard certain (not all) dynamic platforms, and that further research into rotation-mitigation strategies is needed – especially for applications requiring very low size, weight, and power. However, the goal of this work was not to mitigate every noise source that impacts atom interferometry in dynamic environments, but rather to establish error-robust quantum control as a credible approach to mitigating errors (intrinsic and environmental) in cold-atom inertial sensors. Our world-first demonstration opens up an entirely new paradigm of ruggedization designed and implemented in software. We believe that appropriately designed error-robust control protocols could form part of an effective rotation-mitigation strategy, and anticipate that this will be demonstrated in future advances that build upon our pioneering work.

Nevertheless, the two particular noise sources successfully mitigated via this manuscript's software-level robust control techniques are well-established problems facing cold-atom sensors deployed in the field – see, for example, Barrett et al. *Nat. Commun.* 7, 13786 (2016) and Lee et al. *Nat. Commun.* 13, 5131 (2022). Variations in the laser intensity experienced by the atoms – due to both uncontrolled transverse platform motion and the free expansion of the cloud – can be significant for sensing onboard dynamic platforms and even more so for the narrow interferometry beams and large spatial width laser-cooled sources typically used in compact, field-deployable devices. Variations in the laser detuning – due to the atom cloud's velocity spread, or even large amplitude longitudinal vibrations – also significantly degrades the performance of field-deployed devices, which preferentially deploy large velocity-spread atomic sources (in contrast to the much colder sources readily achievable in large, laboratory-based devices).

There are no hardware-level techniques for mitigating degradation caused by the particular error sources studied in this manuscript. In contrast, rotations can be mitigated to some degree through established hardware-level solutions, such as tip-tilt mirrors (Lan et al. *Phys. Rev. Lett.* 108, 090402, 2012) and active gyro-stabilization platforms (Bidel et al. *Nat. Commun.* 9, 627, 2018).

We have updated the introduction to provide a more complete description of the challenges that devices face leaving the lab, and highlight how our approach fits among others.

“Operation on a moving platform -- necessary for deployment on a ship, aircraft, or spacecraft -- typically degrades measurement sensitivity by many orders of magnitude~\cite{Geiger2011, Barrett2016, Bidel2018, Bidel:2020} due to a variety of physical mechanisms. One example mechanism is rotation-induced Coriolis phase shifts \cite{Louchet-Chauvet2011}, which can be mitigated through established hardware solutions such as tip-tilt mirrors \cite{Lan2012} and active gyro-stabilization platforms \cite{Bidel2018}. Another example is variations in the laser intensity experienced by the atoms arising from relative motion between the free-falling atoms and the fixed laser beams \cite{Barrett2016, Lee2022}, which cause errors in the atom-light coupling that degrade the quality of the beamsplitters and mirrors.”

Specific remarks :

The manuscript is well written and could be easily resubmitted to another journal without too much to change.

Mitigation of laser intensity fluctuations

- The authors mention “a constant lateral acceleration of magnitude ~ 1 g is sufficient to push an atom initially at rest in the center of the beam”.

In the context of onboard application, it would be interesting to know what kind of typical environment give rise to 1g lateral acceleration ?

We have expanded our discussion of this point in the “Mitigation of laser intensity fluctuations” section to provide more context:

“The magnitude of constant lateral acceleration required to move an atom initially at rest in the centre of a Gaussian beam with a $1/e^2$ radius of w to a region with 20% lower intensity at the time of the final interferometer pulse is $\approx 0.17w/T^2$. For context, this is ~ 1 g for our beam radius ($w=5$ mm) and maximum interrogation time ($T=10$ ms). Transverse accelerations of this magnitude can occur under relatively benign conditions; for example, any strapdown atom interferometer with three orthogonal measurement axes will experience at least 1 g transverse acceleration if one of the beams is oriented at 90° to local gravity. They can also be induced by platform accelerations in onboard applications, for example due to the sudden turbulence or banking of an aircraft, or during the motion of a marine vessel in moderate to rough seas (e.g. 5 on the Beaufort scale).”

Robust measurement of applied platform acceleration

- Surprisingly 100 μ g vertical acceleration is very small (in comparison with 1 g transverse acceleration). One would have expected the atom accelerometer to be tested with a much broader acceleration magnitude (extending multiple fringe periods) such as the one encountered on references [13-14] for example with typical acceleration values linked to a

representative case-study in terms of acceleration magnitude and frequency.
Can the authors explain why they chose this acceleration magnitude ?

We have added the following text to the section on applied platform acceleration to explain our choice of acceleration magnitude:

"By considering vertical accelerations in the $\sim 100 \mu\text{g}$ range, our interferometric measurements remain within one fringe of the interferometer for interrogation times of relevance to mobile interferometers ($\sim 10 \text{ ms}$). Remaining within one fringe means we can avoid fringe ambiguity without, for example, requiring sensor fusion with a classical co-sensor~\cite{Wang:2023} or implementing schemes to extend the dynamic range \cite{Yankelev2020}."

We are enthusiastic about demonstrating the effectiveness of our error-robust pulses for larger acceleration magnitudes, and intend to do so in future work. However, operating within the dynamic range of a single fringe is more than sufficient to determine the key utility of our approach. Maintaining linearity and accuracy in the presence of noise in the small-signal regime is a necessary (if not sufficient) requirement for useful quantum inertial sensing in mobile applications. Considering vertical accelerations up to $\sim 100 \mu\text{g}$ in magnitude was sufficient to verify that (a) this linearity and accuracy is maintained for our error-robust pulses, (b) our error-robust pulses enable precise and accurate measurements of applied platform accelerations, and (c) this remains the case in the presence of applied intensity noise, whereas the performance of the Gaussian pulse interferometer is significantly degraded.

Methods :

- Atom selection in $m_f=0$

It is not so often that laser light is used to pump the atoms in $m_f=0$.

Why not using Micro-Wave pulse selection ?

The optical pumping scheme described in the Experimental Methods section was sufficient to prepare most atoms in the $m_f=0$ substate. Importantly, this choice does not influence our findings.

- Only one AOM is used to generate the lattice for Bragg pulses.

If I understand well, then two lattices with opposite velocities are generated?

How do you cope with the parasitic lattice ?

We explicitly address this in the experimental methods section:

"Although all the interferometry light shares a common polarization in this configuration, upon retro-reflection only a single Bragg lattice remains resonant in the Doppler-shifted atomic rest frame, with all other transitions being off-resonant following $\sim 20 \text{ ms}$ of freefall."

- Optimal pulse sequence

For optimal pulse, the phase is varied by radians inside the pulse. One would expect somehow that this phase enters the final phase shift calculation in the interferometer.

Could you comment on :

a) the required level of control of the phase

b) how it enters or not the atom interferometer phase shift ?

By design, our optimization procedure aims to produce pulses that impart zero *net* phase shift between the two interferometer output modes. All three pulses are optimized using phase-sensitive cost functions which depend on each pulse's contribution to the final interferometer phase. For example, the mirror cost (Equation 1) is zero only if the operation of the optimized mirror pulse is identical to the operation of an ideal mirror that introduces a constant phase shift for all atoms (Equation 2). Similarly, the cost function used to optimize a pair of beamsplitters is zero only if these pulses introduce no net phase shift in the interferometer fringe. This is captured by Equation 4. Therefore, although the phase varies by radians during each pulse, there is no net phase shift to the output fringe (assuming a cost of zero is reached for each pulse during the optimization).

To address the specific queries (a) and (b) above, we have added the following discussion on the size and potential impact of intra-pulse phase variations into the Methods section:

"Intra-pulse phase variations can in principle introduce spurious phase shifts to the interference fringes. If the atoms do not see the exact waveform (phase, intensity, and frequency profile) obtained from the optimization, then the fidelity of the pulse is lowered, introducing systematic phase shifts into the measurement.

We quantified this in numerical simulation by applying Gaussian phase noise to each ($\SI{1}{\micro s}$) piecewise constant time segment of our light pulse sequence and calculating the standard deviation of the resulting interferometer phase. We found that Gaussian phase noise with $\sigma = \SI{0.1}{\milli rad}$ results in a maximum interferometer phase error of 0.12 and $\SI{0.19}{\milli rad}$ for Gaussian and error-robust pulses, respectively. This is well below the measurement noise on our fringes.

Phase resolution of arbitrary wave synthesis depends on many factors, including phase stability of the oscillator and finite sample rate and amplitude resolution. Many of these result in a synthesized phase error that is negligible compared to the phase uncertainties we quantify here. Nevertheless, as a concrete example for the above result, one could consider a per segment phase error of $\SI{0.1}{\milli rad}$ (e.g. a gross upper bound of the effect of 16-bit amplitude digitization of our AWG)."

Finally, we note that we do not observe any systematic phase shifts that we can attribute to imperfectly realized phase profiles and all biases are bounded to within

6 μ g (see change to CONCLUSIONS AND OUTLOOK section made in response to Reviewer #1, point 11).

Reviewer #3

This paper reports on the application of robust quantum control to a Bragg atom interferometer for mobile sensing. Mobile sensing based on atom interferometry is highly promising and has attracted considerable interest. However, operating a mobile atom interferometer introduces significant challenges, as platform motion tends to dramatically degrade the measurement signal. For example, as the authors note, lateral platform motion can cause the atoms to experience temporally varying laser intensity.

The authors numerically design optimized pulses tailored to be robust against relevant noise sources using a suite of optimization tools that their company (Q-CTRL) has developed. They experimentally demonstrate that incorporating these pulses into their atom interferometer offers order of magnitude level improvements in robustness against various types of noise. Critically, they verify that scale factor stability is maintained. This is an excellent achievement. First, in only one other circumstance has robust quantum control been experimentally shown to improve an atom interferometer (ref. 30). Notably, ref. 30 used Raman pulses, which are much simpler than Bragg pulses due to the multilevel structure of the Bragg interaction, for which many momentum states need to be included in the Hamiltonian. As the authors outline, Bragg interferometers offer important advantages, and the experimental demonstration of a benefit from robust control applied to the complex Bragg system is impressive. Second, never before has anyone demonstrated the benefit of robust control to an actual atom interferometry based inertial sensor.

I expect that this paper will have a broad impact. As the authors describe, mobile sensing based on atom interferometry has a wide range of applications of interest to multiple disciplines. The work reported here will likely inspire others working in this field to adopt robust control techniques. The paper is well-written and can be clearly followed by the reader. For the reasons stated above, I believe that this paper would be a valuable contribution to Nature Communications, provided that the questions and comments below are addressed.

-In Fig. 1c, the amplitude of the optimized pulse appears to be approximately, but not exactly, symmetric in time about the midpoint of the pulse. Could the authors comment on this, and whether there is any reason for this qualitative feature?

We have added the following discussion of pulse symmetry into the Methods section "Design of error robust Bragg pulses":

"Although no constraint on pulse symmetry is applied during our optimization, we find that many optimized pulses have symmetric or almost symmetric waveforms. A temporally symmetric pulse has a symmetric response in frequency space: the target state excitation probability is identical for an atom which is equally positively or negatively detuned from the resonant frequency. Since we optimize pulses for robustness against symmetric momentum distributions (and hence symmetric

two-photon detuning distributions), we speculate that the optimizer finds symmetric solutions because they naturally satisfy this robustness criterion. Similar symmetries are also observed in error-robust pulse design for NMR applications \cite{Kobzar2012}.”

-What motivated the particular choice of third order Bragg diffraction, as opposed to higher (or lower) order? Did the authors try other diffraction orders?

Our choice of third-order diffraction was partly motivated by a desire to demonstrate our control solutions in a large-momentum-transfer (LMT) regime. LMT is potentially of huge benefit to portable sensors with stringent volume and size constraints, since it can improve sensitivity while lowering the required interferometric drop time. We did not test other diffraction orders in this work.

Third-order transitions were a practical compromise between our maximum available laser power and the finite momentum resolution of our detection (constrained by our initial cloud size, available drop time, and ability to cleanly separate each momentum state using our final outcoupling Bloch pulses). Although we could have achieved higher momentum transfer with optimized pulses, we would not have been able to produce equivalent order Gaussian pulses for comparison. Extending our error-robust light sequences to higher Bragg orders is certainly of interest to us, and is the subject of future work. We have therefore added the following to the CONCLUSIONS and OUTLOOK section:

“Future work will ... extend our application of error-robust control to interferometers with larger momentum splittings. Reaching larger momentum orders will likely require higher laser powers, tailored concatenated pulse schemes~\cite{Chiu2011a}, and implementing AC Stark shift mitigation strategies~\cite{Kim2020}. AC Stark shift mitigation should also improve the performance of our error-robust control sequences, especially at longer interrogation times.”

-Could the authors comment on what limited T to 10 ms?

We have added the following to the “Experimental methods” section explaining what limited T to 10ms in our setup:

“We commenced our interferometry sequences $\sim 120\text{ ms}$ after release from the MOT to avoid interrogating a region with a significant magnetic gradient in our chamber. Practically, this means we can only reach interrogation times on the order of $\sim 10\text{ ms}$ while allowing momentum classes to separate appreciably before state detection.”

We emphasize that interrogation times of $\sim 10\text{ms}$ are the relevant regime for operation in dynamic environments due to tight size constraints and limitations imposed by the rotations and vibrations inevitably encountered on moving platforms

(e.g. *Bidel et al., Absolute airborne gravimetry with a cold atom sensor, Journal of Geodesy 94, 20, 2020*).

-It would be useful if the authors could comment on what hardware calibration, if any, was needed for the laser system to faithfully reproduce the robust pulses.

Beyond the employment of a look-up-table to map a desired optical output power to a corresponding AOM drive power at a given frequency (clarified in response to point 12 from Reviewer 1), the process was remarkably calibration-free. The pulse amplitude profile only needs to be scaled to reach the target peak two-photon Rabi frequency, which is quantified via a Rabi oscillation measurement.

-In the paragraph beginning on line 175, it appears that the average improvements offered by robust pulses decreases with increasing T. Do the authors have an explanation for this behavior?

The fringe contrast improvement provided by error-robust pulses is indeed greatest at shorter interrogation times. We suspect this is due to the effect of AC Stark shifts, which are known to cause a T-dependent contrast decay (*Tim Kovachy, PhD thesis, University of Stanford, 2016*) that also depends on the pulse area. Since the error-robust pulses have a larger pulse area than conventional pulses, the contrast decay is worse for these sequences. We do not believe this is a fundamental limitation to the improvement offered by our error-robust pulses, as there exist well-established compensation schemes (e.g. *Kim et al, Opt. Lett. 45, 6555-6558, 2020*) that can mitigate the AC Stark shift in Bragg interferometers. Implementing these mitigation techniques is a focus for future work (as we now point out in the Conclusion and Outlook section).

-The authors study robustness against laser intensity noise that varies up to 20% from pulse to pulse to model transverse motion. For the beam size the authors use and the time scales involved, this corresponds to robustness against impressively large transverse accelerations. What about robustness against similarly large longitudinal accelerations (which one might also expect in inertial navigation applications), which could lead to Doppler shifts that vary from pulse to pulse? The section on applied platform acceleration is relevant to this point, though for smaller longitudinal accelerations as compared to the transverse accelerations needed for 20% intensity variations from pulse to pulse. Do the authors have any thoughts on whether robustness to longitudinal platform accelerations would be maintained for larger longitudinal accelerations or for longitudinal accelerations that significantly vary in time over the course of the interferometer?

We agree that the enhanced detuning robustness of our pulses should enable them to mitigate pulse-to-pulse Doppler shifts arising from larger longitudinal accelerations than conventional pulses. We believe this to be an important application of this technique for onboard applications that we intend to investigate further. We have added a comment to this effect in the CONCLUSIONS AND OUTLOOK section:

“We also expect that the enhanced velocity acceptance of error-robust pulses will mitigate contrast loss caused by longitudinal platform accelerations in onboard applications where these accelerations are large enough to produce appreciable pulse-to-pulse Doppler shifts~\cite{Barrett2016}.”

-The authors have demonstrated robustness against the effects of transverse displacements of the platform. Have they considered whether their methods could be applied to improve robustness against time varying platform tilts, which one might also expect to occur in a mobile sensor? This might be more challenging, since tilts change the direction of the momentum kicks received by the atoms, but I would be curious to hear any thoughts the authors might have on this.

The detuning and intensity-robust pulses we designed and tested in this work would not mitigate contrast loss through the platform tilt-induced Coriolis effect, but may help in regimes where the tilt causes atoms to fall out of the beam. Novel controls would be required to properly mitigate platform tilts for example via pulse timing adjustment informed by feedback from a classical co-sensor.

-In line 249, the authors state that they added stringent band-limits on candidate solutions to reduce population leakage out of the target momentum states. It would be useful if the authors could discuss this point quantitatively, and in greater detail.

We have corrected a typo where the maximum cut-off frequency was stated as Ω_{max} and quoted the correct filter cut-off frequencies used in the design of each pulse in the revised manuscript. We have also provided a quantitative explanation of the filtering procedure in the METHODS subsection “Design of error-robust Bragg pulses”, which we repeat here for the reviewer’s convenience:

“In order to ensure faithful waveform reproduction in hardware and minimize population leakage we also apply a sinc smoothing filter to the control variables $R(t) \equiv \Omega_R(t) \cos[\phi_L(t)]$, $I(t) \equiv \Omega_R(t) \sin[\phi_L(t)]$, and $\delta(t)$, with a maximum cut-off frequency ω_{max} (~ 80 kHz) and ~ 95 kHz for mirrors and beamsplitters, respectively). To apply the filter to a given piecewise-constant control variable $c(t)$, we compute the integral

$$\begin{equation} \int_{-\infty}^{\infty} c(t') \frac{\sin[\omega_{\text{max}}(t-t')]}{\pi(t-t')} dt' = \frac{1}{2\pi} \int_{-\omega_{\text{max}}}^{\omega_{\text{max}}} \tilde{c}(\omega) e^{i\omega t} d\omega, \end{equation}$$

where $\tilde{c}(\omega)$ is the Fourier transform of $c(t)$. This eliminates all frequency components above ω_{max} in $c(t)$. After the filter is applied, the filtered control variable is re-discretized into a piecewise-constant function.”

REVIEWERS' COMMENTS

Reviewer #1 (Remarks to the Author):

The authors have provided detailed responses to my initial comments. All my concerns have been addressed.

I also feel that the authors have responded to the main criticism of Reviewer #2. There is no magic bullet that will solve all the technical challenges associated with bringing atom-interferometric sensors outside the lab. Nevertheless, the authors highlight several issues that can be overcome using software enhancements alone---making this work broadly applicable to the quantum sensing community. I expect they will be adopted by more groups interested in developing these sensors for onboard applications.

The new version of the manuscript is much improved in terms of clarity and detail. I recommend it for publication in Nature Communications.

Reviewer #3 (Remarks to the Author):

I thank the authors for carefully considering the comments of the referees and implementing appropriate changes to address these comments.

There are two particular points on which I would like to comment. The first pertains to the matter of rotations, as referee 2 and I both asked about the challenges of mitigating rotations on a mobile platform. As the authors point out in their response, robust quantum control protocols will not themselves be able to solve this problem. Nevertheless, as the authors correctly note in their response to referee 2, transverse motion of the atoms relative to the laser beam is also a major problem for mobile atom interferometers, which the authors' technique substantially mitigates. In my opinion, it would be too much to ask for a single method to solve all the challenges of sensing with atom interferometers in dynamic environments. Instead, a multifaceted approach is needed. This work makes essential progress by addressing one of the key challenges (transverse motion). Moreover, as I highlighted in my first report, demonstrating for the first time the utility of robust

quantum control for atom interferometry based inertial sensing, especially given the comparatively complex dynamics of Bragg diffraction, is a noteworthy achievement in and of itself that will be of interest to the broader quantum sensing community.

Second, to follow up on the question in my first report about longitudinal accelerations, I agree with the authors that robust control will likely help mitigate the effects of associated Doppler shifts. However, in applications in which longitudinal accelerations are comparable to the 1 g transverse accelerations the authors mention, I expect that the Doppler shifts will be so large that active compensation of the longitudinal motion and/or the laser frequency will be needed to counteract the Doppler shifts. If the authors agree, it would be good for them to note this. If they disagree, then they should explain how they would avoid this problem.

Provided that the point made in the previous paragraph is addressed, I recommend this paper for publication in Nature Communications.

We thank the reviewers for their time and constructive comments, which we address point-by-point below:

Reviewer #1 (Remarks to the Author):

The authors have provided detailed responses to my initial comments. All my concerns have been addressed.

I also feel that the authors have responded to the main criticism of Reviewer #2. There is no magic bullet that will solve all the technical challenges associated with bringing atom-interferometric sensors outside the lab. Nevertheless, the authors highlight several issues that can be overcome using software enhancements alone---making this work broadly applicable to the quantum sensing community. I expect they will be adopted by more groups interested in developing these sensors for onboard applications.

The new version of the manuscript is much improved in terms of clarity and detail. I recommend it for publication in Nature Communications.

We are pleased Reviewer #1 recommends publication of our manuscript.

Reviewer #3 (Remarks to the Author):

I thank the authors for carefully considering the comments of the referees and implementing appropriate changes to address these comments.

There are two particular points on which I would like to comment. The first pertains to the matter of rotations, as referee 2 and I both asked about the challenges of mitigating rotations on a mobile platform. As the authors point out in their response, robust quantum control protocols will not themselves be able to solve this problem. Nevertheless, as the authors correctly note in their response to referee 2, transverse motion of the atoms relative to the laser beam is also a major problem for mobile atom interferometers, which the authors' technique substantially mitigates. In my opinion, it would be too much to ask for a single method to solve all the challenges of sensing with atom interferometers in dynamic environments. Instead, a multifaceted approach is needed. This work makes essential progress by addressing one of the key challenges (transverse motion). Moreover, as I highlighted in my first report, demonstrating for the first time the utility of robust quantum control for atom interferometry based inertial sensing, especially given the comparatively complex dynamics of Bragg diffraction, is a noteworthy achievement in and of itself that will be of interest to the broader quantum sensing community.

Second, to follow up on the question in my first report about longitudinal accelerations, I agree with the authors that robust control will likely help mitigate the effects of associated Doppler shifts. However, in applications in which longitudinal accelerations are comparable to the 1 g transverse accelerations the authors mention, I expect that the Doppler shifts will be so large that active compensation of the longitudinal motion and/or the laser frequency will be needed to counteract the Doppler shifts. If the authors agree, it would be good for

them to note this. If they disagree, then they should explain how they would avoid this problem.

We agree that in situations where the magnitude of longitudinal acceleration is large enough to cause a Doppler shift greater than the velocity acceptance of the error-robust pulses, a technique such as feedforward adjustments to the laser frequency will be required. The magnitude of longitudinal acceleration that can be effectively mitigated by error-robust pulses depends on both the velocity-acceptance of the pulses and the interferometer's interrogation time. For a $T=10$ ms interferometer, a longitudinal acceleration of magnitude 1g would result in a Doppler shift of ~ 500 kHz between the first and final pulses, which is far beyond the velocity acceptance of our error-robust pulses.

We have amended the discussion of vibration compensation in the manuscript, which now reads:

"We also expect that the enhanced velocity acceptance of error-robust pulses would mitigate contrast loss caused by longitudinal platform accelerations in onboard applications, where these accelerations can be large enough to produce appreciable pulse-to-pulse Doppler shifts~\cite{Barrett2016}. The upper bound on the size of longitudinal accelerations that can be mitigated depends upon the interrogation time and pulse velocity acceptance. Consequently, if the acceleration-induced Doppler shift between pulses is larger than the sequence's velocity acceptance, it is likely that active compensation techniques will be required~\cite{Templier:2022, Templier:Thesis} to fully mitigate contrast loss."

Provided that the point made in the previous paragraph is addressed, I recommend this paper for publication in Nature Communications.

We are pleased Reviewer #3 recommends publication of our manuscript, contingent upon a satisfactory response to the point made by the Reviewer above.